# Handling Distribution Shifts on Graphs: An Invariance Perspective

**Qitian Wu**[*]
Shanghai Jiao Tong University
echo740@sjtu.edu.cn

**Hengrui Zhang**[*]
University of Illinois at Chicago
hzhan55@uic.edu

**Junchi Yan**[†]
Shanghai Jiao Tong University
yanjunchi@sjtu.edu.cn

**David Wipf**
Amazon
daviwipf@amazon.com

## Abstract

There is increasing evidence suggesting neural networks' sensitivity to distribution shifts, so that research on out-of-distribution (OOD) generalization comes into the spotlight. Nonetheless, current endeavors mostly focus on Euclidean data, and its formulation for graph-structured data is not clear and remains under-explored, given two-fold fundamental challenges: 1) the inter-connection among nodes in one graph, which induces non-IID generation of data points even under the same environment, and 2) the structural information in the input graph, which is also informative for prediction. In this paper, we formulate the OOD problem on graphs and develop a new invariant learning approach, Explore-to-Extrapolate Risk Minimization (EERM), that facilitates graph neural networks to leverage invariance principles for prediction. EERM resorts to multiple context explorers (specified as graph structure editers in our case) that are adversarially trained to maximize the variance of risks from multiple virtual environments. Such a design enables the model to extrapolate from a single observed environment which is the common case for node-level prediction. We prove the validity of our method by theoretically showing its guarantee of a valid OOD solution and further demonstrate its power on various real-world datasets for handling distribution shifts from artificial spurious features, cross-domain transfers and dynamic graph evolution[1].

## 1 Introduction

As the demand for handling in-the-wild unseen instances draws increasing concerns, out-of-distribution (OOD) generalization (Mansour et al., 2009; Blanchard et al., 2011; Muandet et al., 2013; Gong et al., 2016) occupies a central role in the ML community. Yet, recent evidence suggests that deep neural networks can be sensitive to distribution shifts, exhibiting unsatisfactory performance within new environments, e.g., Beery et al. (2018); Su et al. (2019); Recht et al. (2019); Mancini et al. (2020). A more concerning example is that a model for COVID-19 detection exploits undesired 'shortcuts' from data sources (e.g., hospitals) to boost training accuracy (DeGrave et al., 2021).

Recent studies of the OOD generalization problem like Rojas-Carulla et al. (2018); Bühlmann (2018); Gong et al. (2016); Arjovsky et al. (2019) treat the cause of distribution shifts between training and testing data as a potential unknown environmental variable $\mathbf{e}$. Assuming that the goal is to predict target label $\mathbf{y}$ given associated input $\mathbf{x}$, the environmental variable would impact the underlying data generating distribution $p(\mathbf{x}, \mathbf{y}|\mathbf{e}) = p(\mathbf{x}|\mathbf{e})p(\mathbf{y}|\mathbf{x}, \mathbf{e})$. With $\mathcal{E}$ as the support of environments, $f(\cdot)$ as a prediction model and $l(\cdot, \cdot)$ as a loss function, the OOD problem could be formally represented as

$$\min_f \max_{e \in \mathcal{E}} \mathbb{E}_{(\mathbf{x},y) \sim p(\mathbf{x}, \mathbf{y}|\mathbf{e}=e)}[l(f(\mathbf{x}), y)|e]. \tag{1}$$

---

[*]This work was done during authors' internship at AWS Shanghai AI Lab.

[†]Corresponding author is Junchi Yan. The SJTU authors are also with MoE Key Lab of Artificial Intelligence, Shanghai Jiao Tong University.

[1]The implementation is public available at https://github.com/qitianwu/GraphOOD-EERM.

Such a problem is hard to solve since the observations in training data cannot cover all the environments in practice. Namely, the actual demand is to generalize a model trained with data from $p(\mathbf{x}, \mathbf{y}|\mathbf{e} = e_1)$ to new data from $p(\mathbf{x}, \mathbf{y}|\mathbf{e} = e_2)$. Recent research opens a new possibility via learning domain-invariant models (Arjovsky et al., 2019) under a cornerstone data-generating assumption: there exists a portion of information in $\mathbf{x}$ that is invariant for prediction on $\mathbf{y}$ across different environments. Based on this, the key idea is to learn a *equipredictive* representation model $h$ that gives rise to equal conditional distribution $p(\mathbf{y}|h(\mathbf{x}), \mathbf{e} = e)$ for $\forall e \in \mathcal{E}$. The implication is that such a representation $h(\mathbf{x})$ will bring up equally (optimal) performance for a downstream classifier under arbitrary environments. The model $\hat{p}(\mathbf{y}|\mathbf{x})$ with such a property is called as invariant model/predictor. Several up-to-date studies develop new objective designs and algorithms for learning invariant models, showing promising power for tackling OOD generalization (Chang et al., 2020; Ahuja et al., 2020; Krueger et al., 2021; Liu et al., 2021; Creager et al., 2021; Koyama & Yamaguchi, 2021).

While the OOD problem is extensively explored on Euclidean data (e.g., images), there are few existing works investigating the problem concerning graph-structured data, despite that distribution shifts widely exist in real-world graphs. For instance, in citation networks, the distributions for paper citations (the input) and subject areas/topics (the label) would go through significant change as time goes by (Hu et al., 2020). In social networks, the distributions for users' friendships (the input) and their activity (the label) would highly depend on when/where the networks are collected (Fakhraei et al., 2015). In financial networks (Pareja et al., 2020), the payment flows between transactions (the input) and the appearance of illicit transactions (the label) would have strong correlation with some external contextual factors (like time and market). In these cases, neural models built on graph-structured data, particularly, Graph Neural Networks (GNNs) which are the common choice, need to effectively deal with OOD data during test time. Moreover, as GNNs have become popular and easy-to-implement tools for modeling relational structures in broad AI areas (vision, texts, audio, etc.), enhancing its robustness to distribution shifts is a pain point for building general AI systems, especially applied to high-stake applications like autonomous driving (Dai & Gool, 2018), medical diagnosis (AlBadawy et al., 2018), criminal justice (Berk et al., 2018), etc.

Nonetheless, compared with images or texts, graph-structured data has two fundamental differences. First, many graph-related problems (like the situations mentioned above) involve prediction tasks for each individual node, in which case the data points are inter-connected via graph structure that induces non-independent and non-identically distributed nature in data generation even within the same environment. Second, apart from node features, the structural information also plays a role for prediction and would affect how the model generalizes under environment variation. These differences bring up unique technical challenges for handling distribution shifts on graphs.

In this paper, we endeavor to 1) formulate the OOD problem for node-level tasks on graphs, 2) develop a new learning approach based on an invariance principle, 3) provide theoretical results to dissect its rationale, and 4) design comprehensive experiments to show its practical efficacy. Concretely:

**1.** To accommodate the non-IID nature of nodes in a graph, we fragment a graph into a set of ego-graphs for centered nodes and decompose the data-generating process into: 1) sampling a whole input graph and 2) sampling each node's label conditioned on ego-graph. Based on this, we can inherit the spirit of Eq. 1 to formulate the OOD problem for node-level tasks over graphs (see Section 2.1).

**2.** To account for structural information, we extend the invariance principle with recursive computation on the induced BFS trees of ego-graphs. Then, for out-of-distribution generalization on graphs, we devise a new learning approach, entitled *Explore-to-Extrapolate Risk Minimization*, that aims GNNs at minimizing the mean and variance of risks from multiple environments that are simulated by adversarial context generators (i.e., graph editers), as shown in Fig. 1(a) (see Section 2.2 and 3).

**3.** To shed more insights on the rationales of the proposed approach and its relationship with the formulated OOD problem, we prove that our objective can guarantee a valid solution for the formulated OOD problem given some mild conditions and furthermore, an upper bound on the OOD error can be effectively controlled when minimizing the training error (see Section 4).

**4.** To evaluate the approach, we design a comprehensive set of experiments on diverse real-world node-level prediction datasets that entail distribution shifts from artificial spurious features, cross-domain transfers and dynamic graph evolution. We also apply our approach to distinct GNN backbones (GCN, GAT, GraphSAGE, GCNII and GPRGNN), and the results show that it consistently outperforms standard empirical risk minimization with promising improvements on OOD data (see Section 5).

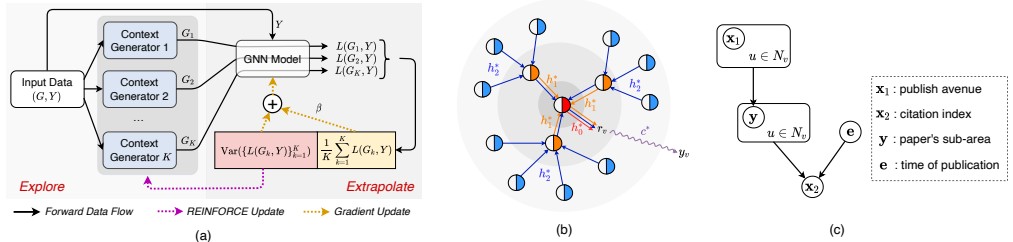

Figure 1: (a) The proposed approach *Explore-to-Extrapolate Risk Minimization* which entails $K$ context generators that generate graph data of different (virtual) environments based on input data from a single (real) environment. The GNN model is updated via gradient descent to minimize a weighted combination of mean and variance of risks from different environments, while the context generators are updated via REINFORCE to maximize the variance loss. (b) Illustration for our Assumption 1. (c) The dependence among variables in the motivating example in Section 3.1.

## 2 PROBLEM FORMULATION

In this section, we present our formulation for the OOD problem on graphs. All the random variables are denoted as bold letters while the corresponding realizations are denoted as thin letters.

### 2.1 OUT-OF-DISTRIBUTION PROBLEM FOR GRAPH-STRUCTURED DATA

An input graph $G = (A, X)$ contains two-fold information[2]: an adjacency matrix $A = \{a_{vu}|v, u \in V\}$ and node features $X = \{x_v|v \in V\}$ where $V$ denotes node set. Apart from these, each node in the graph has a label, which can be represented as a vector $Y = \{y_v|v \in V\}$. We define $\mathbf{G}$ as a random variable of input graphs and $\mathbf{Y}$ as a random variable of node label vectors. Such a definition takes a global view and treat the input graph as a whole. Based on this, one can adapt the definition of general OOD problem Eq. 1 via instantiating the input as $\mathbf{G}$ and the target as $\mathbf{Y}$, and then the data generation can be characterized as $p(\mathbf{G}, \mathbf{Y}|\mathbf{e}) = p(\mathbf{G}|\mathbf{e})p(\mathbf{Y}|\mathbf{G}, \mathbf{e})$ where $\mathbf{e}$ is a random variable of environments that is a latent variable and impacts data distribution.

However, the above definition makes little sense in node-level problems where in most cases there is a single input graph that contains a massive number of nodes. To make the problem-solving reasonable, we instead take a local view and investigate each node's ego-graph that has influence on the centered node. Assume $\mathbf{v}$ as a random variable of nodes. We define node $v$'s $L$-hop neighbors as $N_v$ (where $L$ is an arbitrary integer) and the nodes in $N_v$ form an ego-graph $G_v$ which consists of a (local) node feature matrix $X_v = \{x_u|u \in N_v\}$ and a (local) adjacency matrix $A_v = \{a_{uw}|u, w \in N_v\}$. Use $\mathbf{G_v}$ as a random variable of ego-graphs[3] whose realization is $G_v = (A_v, X_v)$. Besides, we define $\mathbf{y}$ as a random variable of node labels. In this way, we can fragment a whole graph as a set of instances $\{(G_v, y_v)\}_{v \in V}$ where $G_v$ denotes an input and $y_v$ is a target. Notice that the ego-graph can be seen as a Markov blanket for the centered node, so the conditional distribution $p(\mathbf{Y}|\mathbf{G}, \mathbf{e})$ can be decomposed as a product of $|V|$ independent and identical marginal distributions $p(\mathbf{y}|\mathbf{G_v}, \mathbf{e})$.

Therefore, the data generation of $\{(G_v, y_v)\}_{v \in V}$ from a distribution $p(\mathbf{G}, \mathbf{Y}|\mathbf{e})$ can be considered as a two-step procedure: 1) the entire input graph is generated via $G \sim p(\mathbf{G}|\mathbf{e})$ which can then be fragmented into a set of ego-graphs $\{G_v\}_{v \in V}$; 2) each node's label is generated via $y \sim p(\mathbf{y}|\mathbf{G_v} = G_v, \mathbf{e})$. Then the OOD node-level prediction problem can be formulated as: given training data $\{G_v, y_v\}_{v \in V}$ from $p(\mathbf{G}, \mathbf{Y}|\mathbf{e} = e)$, the model needs to handle testing data $\{G_v, y_v\}_{v \in V'}$ from a new distribution $p(\mathbf{G}, \mathbf{Y}|\mathbf{e} = e')$. Denote $\mathcal{E}$ as the support of environments, $f$ as a predictor model with $\hat{y} = f(G_v)$ and $l(\cdot, \cdot)$ as a loss function. More formally, the OOD problem can be written as:

$$\min_f \max_{e \in \mathcal{E}} \mathbb{E}_{G \sim p(\mathbf{G}|\mathbf{e}=e)} \left[ \frac{1}{|V|} \sum_{v \in V} \mathbb{E}_{y \sim p(\mathbf{y}|\mathbf{G_v}=G_v, \mathbf{e}=e)}[l(f(G_v), y)] \right]. \tag{2}$$

We remark that the first-step sampling $G \sim p(\mathbf{G}|\mathbf{e} = e)$ can be ignored since in most cases one only has a single input graph in the context of node-level prediction tasks.

---

[2]Our formulation and method can be trivially extended to cover edge features which we omit here for brevity.

[3]We use a subscript $\mathbf{v}$ here to remind that it is an ego-graph from the view of a target node.

## 2.2 INVARIANT FEATURES FOR NODE-LEVEL PREDICTION ON GRAPHS

To solve the OOD problem Eq. 2 is impossible without any prior domain knowledge or structural assumptions since one only has access to data from limited environments in the training set. Recent studies (Rojas-Carulla et al., 2018; Arjovsky et al., 2019) propose to learn invariant predictor models which resorts to an assumption for data-generating process: the input instance contains a portion of features (i.e., invariant features) that 1) contributes to sufficient predictive information for the target and 2) gives rise to equally (optimal) performance of the downstream classifier across environments.

With our definition in Section 2.1, for node-level prediction on graphs, each input instance is an ego-graph $G_v$ with target label $y_v$. It seems not straightforward for *how to define invariant features on graphs* given two observations: 1) the ego-graph possesses a hierarchical structure for associated nodes (i.e., $G_v$ induces a BFS tree rooted at $v$ where the $l$-th layer contains the $l$-order neighbored nodes $N_v^{(l)}$) and 2) the nodes in each layer are permutation-invariant and variable-length. Inspired by WL test Weisfeiler & Lehman (1968), we extend the invariance assumption (Rojas-Carulla et al., 2018; Gong et al., 2016; Arjovsky et al., 2019) to accommodate structural information in graph data:

**Assumption 1.** *(Invariance Property) Assume input feature dimension as $d_0$. There exists a sequence of (non-linear) functions $\{h_l^*\}_{l=0}^L$ where $h_l^* : \mathbb{R}^{d_0} \to \mathbb{R}^d$ and a permutation-invariant function $\Gamma : \mathbb{R}^{d^m} \to \mathbb{R}^d$, which gives a node-level readout $r_v = r_v^{(L)}$ that is calculated in a recursive way: $r_u^{(l)} = \Gamma\{r_w^{(l-1)}|w \in N_u^{(1)} \cup \{u\}\}$ for $l = 1, \cdots, L$ and $r_u^{(0)} = h_l^*(x_u)$ if $u \in N_v^{(l)}$. Denote $\mathbf{r}$ as a random variable of $r_v$ and it satisfies 1) (Invariance condition): $p(\mathbf{y}|\mathbf{r}, \mathbf{e}) = p(\mathbf{y}|\mathbf{r})$, and 2) (Sufficiency condition): $\mathbf{y} = c^*(\mathbf{r}) + \mathbf{n}$, where $c^*$ is a non-linear function, $\mathbf{n}$ is an independent noise.*

A more intuitive illustration for the above computation is presented in Fig. 1(b). The node-level readout $r_v$ aggregates the information from neighbored nodes recursively along the structures of BFS tree given by $G_v$. Essentially, the above definition assumes that in each layer the neighbored nodes contain a portion of causal features that contribute to stable prediction for $\mathbf{y}$ across different $\mathbf{e}$. Such a definition possesses two merits: 1) the (non-linear) transformation $h_l^*$ can be different across layers, and 2) for arbitrary node $u$ in the original graph $G$, its causal effect on distinct centered nodes $v$ could be different dependent on its relative position in the ego-graph $G_v$. Therefore, this formulation gives rise to enough flexibility and capacity for modeling on graph data.

## 3 METHODOLOGY

We next present our solution for the challenging OOD problem. Before going into the formal method, we first introduce a motivating example based on Assumption 1 to provide some high-level intuition.

### 3.1 MOTIVATING EXAMPLE

We consider a linear toy example and assume 1-layer graph convolution for illustration. Namely, the ego-graph $G_v$ (and $N_v$) only contains the centered node and its 1-hop neighbors. We simplify the $h^*$ and $c^*$ in Assumption 1 as identity mappings and instantiate $\Gamma$ as a mean pooling function. Then we assume 2-dim node features $x_v = [x_v^1, x_v^2]$ and

$$y_v = \frac{1}{|N_v|} \sum_{u \in N_v} x_u^1 + n_v^1, \quad x_v^2 = \frac{1}{|N_v|} \sum_{u \in N_v} y_u + n_v^2 + \epsilon, \tag{3}$$

where $n_v^1$ and $n_v^2$ are independent standard normal noise and $\epsilon$ is a random variable with zero mean and non-zero variance dependent on environment $e$. In Fig. 1(c) we show the dependency among these random variables in a graphical representation and instantiate them in an example of citation networks, where a paper's published avenue is an invariant feature for predicting the paper's sub-area while its citation index (a spurious feature) is affected by both the label and the environment.

Based on this, we consider a vanilla GCN as the predictor model $\hat{y}_v = \frac{1}{|N_v|} \sum_{u \in N_v} \theta_1 x_u^1 + \theta_2 x_u^2$. Then the ideal solution for the predictor model is $[\theta_1, \theta_2] = [1, 0]$. This indicates that the GCN identifies the invariant feature, i.e., $x_v^1$ insensitive to environment changes. However, here we show a negative result when using standard empirical risk minimization.

**Proposition 1.** *Let the risk under environment $e$ be $R(e) = \frac{1}{|V|} \sum_{v \in V} \mathbb{E}_{\mathbf{y}|\mathbf{G_v}=G_v}[\|\hat{y}_v - y_v\|_2^2]$. The unique optimal solution for objective $\min_\theta \mathbb{E}_\mathbf{e}[R(e)]$ would be $[\theta_1, \theta_2] = [\frac{1+\sigma_e^2}{2+\sigma_e^2}, \frac{1}{2+\sigma_e^2}]$ where $\sigma_e > 0$ denotes the standard deviation of $\epsilon$ across environments.*

This indicates that directly minimizing the expectation of risks across environments would inevitably lead the model to rely on spurious correlation ($x_v^2$ depends on environments). Also, such a reliance would be strengthened with smaller $\sigma_e$, i.e., when there is less uncertainty for the effect from environments. To mitigate the issue, fortunately, we can prove another result that implies a new objective as a sufficient condition for the ideal solution.

**Proposition 2.** *The objective* $\min_\theta \mathbb{V}_e[R(e)]$ *reaches the optimum if and only if* $[\theta_1, \theta_2] = [1, 0]$.

The new objective tackles the variance across environments and guarantees the desirable solution. The enlightenment is that if the model yields equal performance on different $e$'s, it would manage to leverage the invariant features, which motivates us to devise a new objective for solving Eq. 2.

### 3.2 STABLE LEARNING WITH EXPLORE-TO-EXTRAPOLATE RISK MINIMIZATION

We now return to the general case where we have $\{(G_v, y_v)\}$ for training and leverage a GNN model as the predictor: $\hat{y}_v = f_\theta(G_v)$. The intuition in Section 3.1 implies a new learning objective:

$$\min_\theta \mathbb{V}_{\mathbf{e}}[L(G^e, Y^e; \theta)] + \beta \mathbb{E}_{\mathbf{e}}[L(G^e, Y^e; \theta)], \tag{4}$$

where $L(G^e, Y^e; \theta) = \frac{1}{|V_e|} \sum_{v \in V_e} l(f_\theta(G_v^e), y_v^e)$ and $\beta$ is a trading hyper-parameter. If we have training graphs from a sufficient number of environments $\mathcal{E}_{tr} = \{e\}$ and the correspondence of each graph to a specific $e$, i.e., $\{G^e, Y^e\}_{e \in \mathcal{E}_{tr}}$ which induces $\{\{G_v^e, y_v^e\}_{v \in V_e} : e \in \mathcal{E}_{tr}\}$, we can use the empirical estimation with risks from different environments to handle Eq. 4 in practice, as is done by the Risk Extrapolation (REX) approach (Krueger et al., 2021). Unfortunately, as mentioned before, for node-level tasks on graphs, the training data is often a single graph (without any correspondence of nodes to environments), and hence, one only has training data from a single environment. Exceptions are some multi-graph scenarios where one can assume each graph is from an environment, but there are still a very limited number of training graphs (e.g., less than five). The objective Eq. 4 would require data from diverse environments to enable the model for desirable extrapolation. To detour such a dilemma, we introduce $K$ auxiliary context generators $g_{w_k}(G)$ $(k = 1, \cdots, K)$ that aim to generate $K$-fold graph data $\{G^k\}_{k=1}^K$ (which induces $\{\{G_v^k\}_{v \in V} : 1 \leq k \leq K\}$) based on the input one $G$ and mimics training data from different environments. The generators are trained to maximize the variance loss so as to explore the environments and facilitate stable learning of the GNN:

$$\min_\theta \mathrm{Var}(\{L(g_{w_k^*}(G), Y; \theta) : 1 \leq k \leq K\}) + \frac{\beta}{K} \sum_{k=1}^K L(g_{w_k^*}(G), Y; \theta),$$

$$\text{s. t. } [w_1^*, \cdots, w_K^*] = \arg \max_{w_1, \cdots, w_K} \mathrm{Var}(\{L(g_{w_k}(G), Y; \theta) : 1 \leq k \leq K\}), \tag{5}$$

where $L(g_{w_k}(G), Y; \theta) = L(G^k, Y; \theta) = \frac{1}{|V|} \sum_{v \in V} l(f_\theta(G_v^k), y_v)$.

One remaining problem is how to specify $g_{w_k}(G)$. Following recent advances in adversarial robustness on graphs (Xu et al., 2019; Jin et al., 2020), we consider editing graph structures by adding/deleting edges. Assume a Boolean matrix $B^k = \{0, 1\}^{N \times N}$ $(k = 1, \cdots, K)$ and denote the supplement graph of $A$ as $\overline{A} = \mathbf{1}\mathbf{1}^\top - I - A$, where $I$ is an identity matrix. Then the modified graph for view $k$ is $A^k = A + B^k \circ (\overline{A} - A)$ where $\circ$ denotes element-wise product. The optimization for $B^k$ is difficult due to its non-differentiability and one also needs to constrain the modification within a threshold. To handle this, we use policy gradient method REINFORCE, treating graph generation as a decision process and edge editing as actions (see details in Appendix A). We call our approach in Eq. 5 *Explore-to-Extrapolate Risk Minimization* (EERM) and present our training algorithm in Alg. 1.

## 4 THEORETICAL DISCUSSIONS

We next present theoretical analysis to shed insights on the objective and its relationship with our formulated OOD problem in Section 2.1. To begin with, we introduce some building blocks. The GNN model $f$ can be decomposed into an *encoder* $h$ for representation and a *classifier* $c$ for prediction, i.e., $f = c \circ h$ and we have $z_v = h(G_v)$, $\hat{y}_v = c(z_v)$. Besides, we assume $I(\mathbf{x}; \mathbf{y})$ stands for the mutual information between $\mathbf{x}$ and $\mathbf{y}$ and $I(\mathbf{x}; \mathbf{y}|\mathbf{z})$ denotes the conditional mutual information given $\mathbf{z}$. To keep notations simple, we define $p_e(\cdot) = p(\cdot|\mathbf{e} = e)$ and $I_e(\cdot) = I(\cdot|\mathbf{e} = e)$. Another tricky point is that in computation of the KL divergence and mutual information, we require

the samples from the joint distribution $p_e(\mathbf{G}, \mathbf{Y})$, which also results in difficulty for handling data generation of interconnected nodes. Therefore, we again adopt our perspective in Section 2.1 and consider a two-step sampling procedure. Concretely, for any probability function $f_1$, $f_2$ associated with ego-graphs $\mathbf{G_v}$ and node labels $\mathbf{y}$, we define computation for KL divergence as

$$D_{KL}(f_1(\mathbf{G_v}, \mathbf{y}) \| f_2(\mathbf{G_v}, \mathbf{y})) := \mathbb{E}_{G \sim p(\mathbf{G})}\left[ \frac{1}{|V|} \sum_{v \in V} \mathbb{E}_{y_v \sim p(\mathbf{y}|\mathbf{G_v}=G_v)} \left[ \log \frac{f_1(\mathbf{G_v}=G_v, \mathbf{y}=y_v)}{f_2(\mathbf{G_v}=G_v, \mathbf{y}=y_v)} \right] \right].$$
(6)

### 4.1 RELATIONSHIP BETWEEN INVARIANCE PRINCIPLE AND OOD PROBLEM

We will show that the objective Eq. 4 can guarantee a valid solution for OOD problem Eq. 2. To this end, we rely on another assumption for data-generating distribution.

**Assumption 2.** *(Environment Heterogeneity): For $(\mathbf{G_v}, \mathbf{r})$ that satisfies Assumption 1, there exists a random variable $\bar{\mathbf{r}}$ such that $\mathbf{G_v} = m(\mathbf{r}, \bar{\mathbf{r}})$ where $m$ is a functional mapping. We assume that $p(\mathbf{y}|\bar{\mathbf{r}}, \mathbf{e} = e)$ would arbitrarily change across environments $e \in \mathcal{E}$.*

Assumptions 1 and 2 essentially distill two portions of features in input data: one is domain-invariant for prediction and the other contributes to sensitive prediction that depends on environments. The GNN model $f = c \circ h$ induces two model distributions $q(\mathbf{z}|\mathbf{G_v})$ (by the encoder) and $q(\mathbf{y}|\mathbf{z})$ (by the classifier). Based on this, we can dissect the effects of Eq. 4 which indeed forces the representation $\mathbf{z}$ to satisfy the *invariance* and *sufficiency* conditions illustrated in Assumption 1.

**Theorem 1.** *If $q(\mathbf{y}|\mathbf{z})$ is treated as a variational distribution, then 1) minimizing the expectation term in Eq. 4 contributes to $\max_{q(\mathbf{z}|\mathbf{G_v})} I(\mathbf{y}; \mathbf{z})$, i.e., enforcing the sufficiency condition on $\mathbf{z}$ for prediction, and 2) minimizing the variance term in Eq. 4 would play a role for $\min_{q(\mathbf{z}|\mathbf{G_v})} I(\mathbf{y}; \mathbf{e}|\mathbf{z})$, i.e., enforcing the invariance condition $p(\mathbf{y}|\mathbf{z}, \mathbf{e}) = p(\mathbf{y}|\mathbf{z})$.*

Based on these results, we can bridge the gap between the invariance principle and OOD problem.

**Theorem 2.** *Under Assumption 1 and 2, if the GNN encoder $q(\mathbf{z}|\mathbf{G_v})$ satisfies that 1) $I(\mathbf{y}; \mathbf{e}|\mathbf{z}) = 0$ (invariance condition) and 2) $I(\mathbf{y}; \mathbf{z})$ is maximized (sufficiency condition), then the model $f^*$ given by $\mathbb{E}_\mathbf{y}[\mathbf{y}|\mathbf{z}]$ is the solution to OOD problem in Eq. 2.*

The above results imply that the objective Eq. 4 can guarantee a valid solution for the formulated OOD problem on graph-structured data, which serves as a theoretical justification for our approach.

### 4.2 INFORMATION-THEORETIC ERROR FOR OOD GENERALIZATION

We proceed to analyze the OOD generalization error given by our learning approach. Recall that we assume training data from $p(\mathbf{G}, \mathbf{Y}|\mathbf{e} = e)$ and testing data from $p(\mathbf{G}, \mathbf{Y}|\mathbf{e} = e')$. Following similar spirits of Federici et al. (2021), the training error and OOD generalization error can be respectively measured by $D_{KL}(p_e(\mathbf{y}|\mathbf{G_v})\|q(\mathbf{y}|\mathbf{G_v}))$ and $D_{KL}(p_{e'}(\mathbf{y}|\mathbf{G_v})\|q(\mathbf{y}|\mathbf{G_v}))$ which can be calculated based on our definition in Eq. 6. Based on Theorem 1, we can arrive at the following theorem which reveals the effect of Eq. 4 that contributes to tightening the bound for the OOD error.

**Theorem 3.** *Optimizing Eq. 4 with training data can minimize the upper bound for $D_{KL}(p_{e'}(\mathbf{y}|\mathbf{G_v})\|q(\mathbf{y}|\mathbf{G_v})$ on condition that $I_{e'}(\mathbf{G_v}; \mathbf{y}|\mathbf{z}) = I_e(\mathbf{G_v}; \mathbf{y}|\mathbf{z})$.*

The condition can be satisfied once $\mathbf{z}$ is a sufficient representation across environments. Therefore, we have proven that the new objective could help to reduce the generalization error on out-of-distribution data and indeed enhance GNN model's power for in-the-wild extrapolation.

## 5 EXPERIMENTS

In this section, we aim to verify the effectiveness and robustness of our approach in a wide variety of tasks reflecting real situations, using different GNN backbones. Table 1 summarizes the information of experimental datasets and evaluation protocols, and we provide more dataset information in Appendix E. We compare our approach EERM with standard empirical risk minimization (ERM). Implementation details are presented in Appendix F. In the following subsections, we will investigate three scenarios that require the model to handle distribution shifts stemming from different causes.

Table 1: Summary of the experimental datasets that entail diverse distribution shifts ("Artificial Transformation" means that we add synthetic spurious features, "Cross-Domain Transfers" means that each graph in the dataset corresponds to distinct domains, "Temporal Evolution" means that the dataset is a dynamic one with evolving nature), different train/val/test splits ("Domain-Level" means splitting by graphs and "Time-Aware" means splitting by time) and the evaluation metrics. In Appendix E we provide more detailed information and discussions on the evaluation protocols.

| Dataset | Distribution Shift | #Nodes | #Edges | #Classes | Train/Val/Test Split | Metric | Adapted From |
|---|---|---|---|---|---|---|---|
| Cora | Artificial Transformation | 2,703 | 5,278 | 10 | Domain-Level | Accuracy | Yang et al. (2016) |
| Amazon-Photo | | 7,650 | 119,081 | 10 | Domain-Level | Accuracy | Shchur et al. (2018) |
| Twitch-explicit | Cross-Domain Transfers | 1,912 - 9,498 | 31,299 - 153,138 | 2 | Domain-Level | ROC-AUC | Rozemberczki et al. (2021) |
| Facebook-100 | | 769 - 41,536 | 16,656 - 1,590,655 | 2 | Domain-Level | Accuracy | Traud et al. (2011) |
| Elliptic | Temporal Evolution | 203,769 | 234,355 | 2 | Time-Aware | F1 Score | Pareja et al. (2020)[1] |
| OGB-Arxiv | | 169,343 | 1,166,243 | 40 | Time-Aware | Accuracy | Hu et al. (2020) |

[1] The original dataset is provided at `https://www.kaggle.com/ellipticco/elliptic-data-set`.

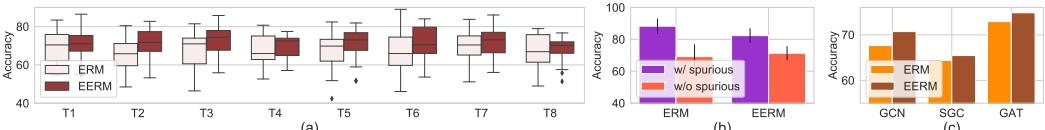

Figure 2: Results on `Cora` with artificial distribution shifts. We run each experiment with 20 trials. (a) The (distribution of) test accuracy of vanilla GCN using our approach for training and using ERM. (b) The (averaged) accuracy on the training set (achieved by the epoch where the highest validation accuracy is achieved) when using all the input node features and removing the spurious ones for inference. (c) The (averaged) test accuracy with different GNNs for data generation.

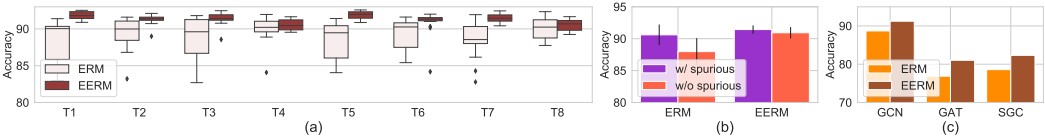

Figure 3: Experiment results on `Amazon-Photo` with artificial distribution shifts.

## 5.1 HANDLING DISTRIBUTION SHIFTS WITH ARTIFICIAL TRANSFORMATION

We first consider artificial distribution shifts based on two public node classification benchmarks `Cora` and `Amazon-Photo`. For each dataset, we adopt two randomly initialized GNNs to 1) generate node labels based on the original node features and 2) generate spurious features based on the node labels and environment id, respectively (See Appendix E.1 for details). We generate 10-fold graph data with distinct environment id's and use 1/1/8 of them for training/validation/testing.

We use a 2-layer vanilla GCN (Kipf & Welling, 2017) as the GNN model. We report results on 8 testing graphs (T1∼ T8) of the two datasets in Fig. 2(a) and 3(a), respectively, where we also adopt 2-layer GCNs for data generation. The results show that our approach EERM consistently outperforms ERM on `Cora` and `Photo`, which suggests the effectiveness of our approach for handling distribution shifts. We also observe that in `Photo`, the performance variances within one graph and across different test graphs are both much lower compared with those in `Cora`. We conjecture the reasons are two-fold. First, there is evidence that in `Cora` the (original) features from adjacent nodes are indeed informative for prediction while in `Photo` this information contributes to negligible gain over merely using centered node's features. Based on this, once the node features are mixed up with invariant and spurious ones, it would be harder for distinguishing them in the former case that relies more on graph convolution.

In Fig. 2(b) and Fig. 3(b), we compare the averaged training accuracy (achieved by the epoch with the highest validation accuracy) given by two approaches when using all the input features and removing the spurious ones for inference (we still use all the features for training in the latter case). As we can see, the performance of ERM drops much more significantly than EERM when we remove the spurious input features, which indicates that the GCN trained with standard ERM indeed exploits spurious features to increase training accuracy while our approach can help to alleviate such an issue and guide the model to focus on invariant features. Furthermore, in Fig. 2(c) and Fig. 3(c), we compare the test accuracy averaged on eight graphs when using different GNNs e.g. GCN, SGC (Wu et al., 2019) and GAT (Velickovic et al., 2018), for data generation (See Appendix G for more results). The results verify that our approach achieves consistently superior performance in different cases.

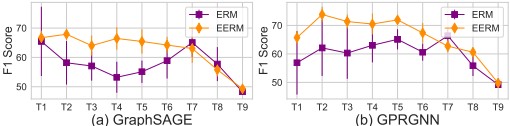

Figure 4: Test ROC-AUC on `Twitch` where we compare different GNN backbones.

Table 2: Test accuracy on `FB-100` where we compare different configurations of training graphs.

| Training graph combination | Penn | | Brown | | Texas | |
|---|---|---|---|---|---|---|
| | ERM | EERM | ERM | EERM | ERM | EERM |
| John Hopkins + Caltech + Amherst | $50.48 \pm 1.09$ | $50.64 \pm 0.25$ | $54.53 \pm 3.93$ | $56.73 \pm 0.23$ | $53.23 \pm 4.49$ | $55.57 \pm 0.75$ |
| Bingham + Duke + Princeton | $50.17 \pm 0.65$ | $50.67 \pm 0.79$ | $50.43 \pm 4.58$ | $52.76 \pm 3.40$ | $50.19 \pm 5.81$ | $53.82 \pm 4.88$ |
| WashU + Brandeis+ Carnegie | $50.83 \pm 0.17$ | $51.52 \pm 0.87$ | $54.61 \pm 4.75$ | $55.15 \pm 3.22$ | $56.25 \pm 0.13$ | $56.12 \pm 0.42$ |

Figure 5: Test F1 score on `Elliptic` where we group graph snapshots into 9 test sets (T1~ T9).

Table 3: Test accuracy on `OGB-Arxiv` with papers in different time intervals for evaluation.

| Method | 2014-2016 | 2016-2018 | 2018-2020 |
|---|---|---|---|
| ERM- SAGE | $42.09 \pm 1.39$ | $39.92 \pm 2.53$ | $36.72 \pm 2.47$ |
| EERM- SAGE | $41.55 \pm 0.68$ | $40.36 \pm 1.29$ | $38.95 \pm 1.57$ |
| ERM- GPR | $47.25 \pm 0.55$ | $45.07 \pm 0.57$ | $41.53 \pm 0.56$ |
| EERM- GPR | $49.88 \pm 0.49$ | $48.59 \pm 0.52$ | $44.88 \pm 0.62$ |

## 5.2 GENERALIZING TO UNSEEN DOMAINS

We proceed to consider another scenario where there are multiple observed graphs in one dataset and a model trained with one graph or a limited number of graphs is expected to generalize to new unseen graphs. The graphs of a dataset share the input feature space and output space and may have different sizes and data distributions since they are collected from different domains. We adopt two public social network datasets `Twitch-Explicit` and `Facebook-100` collected by Lim et al. (2021).

**Training with a Single Graph.** In `Twitch`, we adopt a single graph DE for training, ENGB for validation and the remaining five networks (ES, FR, PTBR, RU, TW) for testing. We follow Lim et al. (2021) and use test ROC-AUC for evaluation. We specify the GNN model as GCN, GAT and a recently proposed model GCNII (Chen et al., 2020a) that can address the over-smoothing of GCN and enable stacking of deep layers. The layer numbers are set as 2 for GCN and GAT and 10 for GCNII. Fig. 4 compares the results on five test graphs, where EERM significantly outperforms ERM in most cases with up to 7.0% improvement on ROC-AUC. The results verify the effectiveness of EERM for generalizing to new graphs from unseen domains.

**Training with Multiple Graphs.** In `FB-100`, we adopt three graphs for training, two graphs for validation and the remaining three for testing. We also follow Lim et al. (2021) and use test accuracy for evaluation. We use GCN as the backbone and compare using different configurations of training graphs, as shown in Table 2. We can see that EERM outperforms ERM on average on all the test graphs with up to 7.2% improvement. Furthermore, EERM maintains the superiority with different training graphs, which also verifies the robustness of our approach w.r.t. training data.

## 5.3 EXTRAPOLATING OVER DYNAMIC DATA

We consider the third scenario where the input data are temporal dynamic graphs and the model is trained with dataset collected at one time and needs to handle newly arrived data in the future. Here are also two sub-cases that correspond to distinct real-world scenarios, as discussed below.

**Handling Dynamic Graph Snapshots.** We adopt a dynamic financial network dataset `Elliptic` (Pareja et al., 2020) that contains dozens of graph snapshots where each node is a Bitcoin transaction and the goal is to detect illicit transactions. We use 5/5/33 snapshots for train/valid/test. Following Pareja et al. (2020) we use F1 score for evaluation. We consider two GNN architectures as the backbone: GraphSAGE (Hamilton et al., 2017) and GPRGNN (Chien et al., 2021) that can adaptively combine information from node features and graph topology. The results are shown in Fig. 5 where we group the test graph snapshots into 9 folds in chronological order. Our approach yields much higher F1 scores than ERM in most cases with averagely 9.6%/10.0% improvements using GraphSAGE/GPR-GNN as backbones. Furthermore, there is an interesting phenomenon that both methods suffer a performance drop after T7. The reason is that this is the time when the dark market shutdown occurred (Pareja et al., 2020). Such an emerging event causes considerable variation to data

distributions that leads to performance degrade for both methods, with ERM suffering more. In fact, the emerging event acts as an external factor which is unpredictable given the limited training data. The results also suggest that how neural models generalize to OOD data depends on the learning approach but its performance limit is dominated by observed data. Nonetheless, our approach contributes to better F1 scores than ERM even in such an extreme case.

**Handling New Nodes in Temporally Augmented Graph.** Citation networks often go through temporal augmentation with new papers published. We adopt `OGB-Arxiv` (Hu et al., 2020) for experiments and enlarge the time difference between training and testing data to introduce distribution shifts: we select papers published before 2011 for training, in-between 2011 and 2014 for validation, and within 2014-2016/2016-2018/2018-2020 for testing. Also different from the original (transductive) setting in Hu et al. (2020), we use the inductive learning setting, i.e., test nodes are strictly unseen during training, which is more akin to practical situations. Table 3 presents the test accuracy and shows that EERM outperforms ERM in five cases out of six. Notably, when using GPRGNN as the backbone, EERM manages to achieve up to $8.1\%$ relative improvement, which shows that EERM is capable of improving GNN model's learning for extrapolating to future data.

## 6 DISCUSSIONS WITH EXISTING WORKS

We compare with some closely related works and highlight our differences. Due to space limit, we defer more discussions on literature review to Appendix B.

**Generalization/Extrapolation on Graphs.** Recent endeavors (Scarselli et al., 2018; Garg et al., 2020; Verma & Zhang, 2019) derive generalization error bounds for GNNs, yet they focus on in-distribution generalization and put little emphasis on distribution shifts, which are the main focus of our work. Furthermore, some up-to-date works explore GNN's extrapolation ability for OOD data, e.g. unseen features/structures (Xu et al., 2021) and varying graph sizes (Yehudai et al., 2021; Bevilacqua et al., 2021). However, they mostly concentrate on graph-level tasks rather than node-level ones (see detailed comparison below). Moreover, some recent works probe into extrapolating features' embeddings (Wu et al., 2021a) or user representations (Wu et al., 2021b) for open-world learning in tabular data or real systems for recommendation and advertisement. These works consider distribution shifts stemming from augmented input space or unseen entities, and the proposed models leverage GNNs as an explicit model that extrapolates to compute representations for new entities based on existing ones. In contrast, our work studies (implicit) distribution shifts behind observed data, and the proposed approach resorts to an implicit mechanism through the lens of invariance principle.

**Node-Level v. s. Graph-Level OOD Generalization.** The two classes of graph-related problems are often treated differently in the literature: node-level tasks target prediction on individual nodes that are non-i.i.d. generated due to the interconnection of graph structure; graph-level tasks treat a graph as an instance for prediction and tacitly assume that all the graph instances are sampled in an i.i.d. manner. The distinct nature of them suggests that they need to be tackled in different ways for OOD generalization purpose. Graph-level problems have straightforward relationship to the general setting (in Eq. 1) since one can treat input graphs as $x$ and graph labels as $y$. Based on this, the data from one environment becomes a set of i.i.d. generated pairs $(x, y)$ and existing approaches for handling general OOD settings could be naturally generalized. Differently, node-level problems, where nodes inter-connected in one graph (e.g., social network) are instances, cannot be handled in this way due to the non-i.i.d. data generation. Our work introduces a new perspective for formulating OOD problems involving prediction for individual nodes, backed up with a concrete approach for problem solving and theoretical guarantees. While our experiments mainly focus on node-level prediction datasets, the proposed method can be applied to graph-level prediction and also extended beyond graph data (like images/texts) for generalization from limited observed environments.

## 7 CONCLUSION

This work targets out-of-distribution generalization for graph-structured data with the focus on node-level problems where the inter-connection of data points hinders trivial extension from existing formulation and methods. To this end, we take a fresh perspective to formulate the problem in a principled way and further develop a new approach for extrapolation from a single environment, backed up with theoretical guarantees. We also design comprehensive experiments to show the practical power of our approach on various real-world datasets with diverse distribution shifts.

## 8 ACKNOWLEDGEMENT

This work was in part supported by National Key Research and Development Program of China (2020AAA0107600), Shanghai Municipal Science and Technology Major Project (2021SHZDZX0102).

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

## A  OPTIMIZATION AND ALGORITHM

We illustrate the details of using policy gradient for optimizing the graph editers in Eq. 5. Concretely, for view $k$, we consider a parameterized matrix $P_k = \{\pi_{nm}^k\}$. For the $n$-th node, the probability for editing the edge between it and the $m$-th node would be $p(a_{nm}^k) = \frac{\exp(\pi_{nm}^k)}{\sum_{m'} \exp(\pi_{nm'}^k)}$. We then sample $s$ actions $\{b_{nm_t}^k\}_{t=1}^s$ from a multinomial distribution $\mathcal{M}(p(a_{n1}^k), \cdots, p(a_{nn}^k))$, which give the non-zero entries in the $n$-th row of $B^k$. The reward function $R(G^k)$ can be defined as the inverse loss. We can use REINFORCE algorithm to optimize the generator with the gradient $\nabla_{w_k} \log p_{w_k}(A^k) R(G^k)$ where $w_k = P_k$ and $p_{w_k}(A^k) = \Pi_n \Pi_{t=1}^s p(b_{nm_t}^k)$. We present the training algorithm in Alg. 1.

---

**Algorithm 1:** Stable Learning for OOD Generalization in Node-Level Prediction on Graphs.

---

1  **INPUT:** training graph data $G = (A, X)$ and $Y$, initialized parameters of GNN $\theta$, initialized parameters of graph editers $w = \{w_k\}$, learning rates $\alpha_g, \alpha_f$.
2  **while** *not converged or maximum epochs not reached* **do**
3      **for** $t = 1, \cdots, T$ **do**
4          Obtain modified graphs $G^k = (A^k, X)$ from graph editer $g_{w_k}$, $k = 1, \cdots, K$;
5          Compute loss $J_1(w) = \mathrm{Var}(\{L(G^k, Y; \theta) : 1 \le k \le K\})$ ;
6          Update $w_k \leftarrow w_k + \alpha_g \nabla_{w_k} \log p_{w_k}(A^k) J_1(w)$, $k = 1, \cdots, K$ ;
7          **if** $t == T$ **then**
8              Compute loss $J_2(\theta) = \mathrm{Var}(\{L(G^k, Y; \theta) : 1 \le k \le K\}) + \frac{\beta}{K} \sum_{k=1}^K L(G^k, Y; \theta)$ ;
9              Update $\theta \leftarrow \theta - \alpha_f \nabla_\theta J_2(\theta)$ ;
10  **OUTPUT:** trained parameters of GNN $\theta^*$.

---

## B  FURTHER RELATED WORKS

### B.1  OUT-OF-DISTRIBUTION GENERALIZATION AND INVARIANT MODELS

Out-of-distribution generalization has drawn extensive attention in the machine learning community. To endow the learning systems with the ability for handling unseen data from new environments, it is natural to learn invariant features under the setting of the causal factorization of physical mechanisms (Schölkopf et al., 2012; Peters et al., 2016). A recent work (Arjovsky et al., 2019) proposes Invariant Risk Minimization (IRM) as a practical solution for OOD problem via invariance principle. Based on this, follow-up works make solid progress in this direction, e.g., with group distributional robust optimization (Sagawa et al., 2019), invariant rationalization (Chang et al., 2020), game theory (Ahuja et al., 2020), etc. Several works attempt to resolve extended settings. For instance, Ahmed et al. (2021) proposes to match the output distribution spaces from different domains via some divergence, while a recent work (Mahajan et al., 2021) also leverages a matching-based algorithm that resorts to shared representations of cross-domain inputs from the same object. Also, Creager et al. (2021) and Liu et al. (2021) point out that in most real situations, one has no access to the correspondence of each data point in the dataset with a specific environment, based on which they propose to estimate the environments as a latent variable.

Krueger et al. (2021) devises Risk Extrapolation (REX) which aims at minimizing the weighted combination of the variance and the mean of risks from multiple environments. Xie et al. (2020) contributes to a similar objective from different theoretical perspective. Also, Koyama & Yamaguchi (2021) extends the spirit of MAML algorithm and arrives at a similar objective form. In our model, we also consider minimization of the combination of variance and mean terms (in Eq. 4) and on top of that we further propose to optimize through a bilevel framework in Eq. 5. Compared to existing works, the differences of EERM are two-folds. First, we do not assume input data from multiple environments and the correspondence between each data point and a specific environment. Instead, our formulation enables learning and extrapolation from a single observed environment. Second, on methodology side, we introduce multiple context generators that aim to generate data of virtual environments in an adversarial manner. Besides, our formulation in this paper focus on prediction tasks for nodes on graphs, where the essential difference, as mentioned before, lies in the

inter-connection of data points in one graph (that corresponds to an environment), which hinders trivial adaption from existing works in the general setting.

## B.2 GRAPH NEURAL NETWORKS AND GENERALIZATION

Another line of research related to us attempts to enhance the generalization ability of graph neural networks via modifying the graph structures. One category of recent works is to learn new graph structures based on the input graph and node features. To improve the generalization power, a common practice is to enforce a certain regularization for the learned graph structures. For example, Jin et al. (2020) proposes to constrain the sparsity and smoothness of graphs via matrix norms and further adopts proximal gradient methods for handling the non-differentiable issue. Chen et al. (2020b); Zhang et al. (2019) also aim to regularize the sparsity and smoothness but differently harness energy function to enforce the constraints. From a different perspective, Xu et al. (2019) attempts to attack the graph topology for improving the model's robustness and proposes to leverage projected gradient descent to make it tractable for the optimization of discrete graph structures. Alternatively, several works focus on pruning the graph networks (Chen et al., 2021) or adaptively sparsifying the structures (Zheng et al., 2020; Hasanzadeh et al., 2020). In this paper, we focus on out-of-distribution generalization and target handling distribution shifts over graphs, which is more difficult than the setting of previous methods that concentrate on in-distribution generalization. On methodology side, with a different training scheme, our context generators aim to generate data of multiple virtual environments and are learned from maximizing the variance of risks of multiple (virtual) environments. Also, one can specify our context generators with other existing graph generation/editing/attacking frameworks as mentioned above, which we leave as future works.

There are some very recent works that endeavor to study the out-of-distribution generalization ability of GNNs for node classification. (Baranwal et al., 2021) introduces contextual stochastic block model that is a mix-up of standard stochastic block model and a Gaussian mixture model with each node class corresponding to a component, based on which the authors show some cases where linear separability can be achieved. (Ma et al., 2021) focuses on subgroup generalization and presents a PAC-Bayesian theoretic framework, while (Zhu et al., 2021) targets distribution shifts between selecting training and testing nodes and proposes new GNN model called Shift-Robust GNNs. By contrast, our work possesses the following distinct technical contributions. First, we formulate OOD problem for node-level tasks in a general manner without any assumption on specific distribution forms or the way for generation of graph structures (our Assumption 1 and 2 mainly focus on the existence of causal features and spurious features in data generation across different environments). Second, our proposed model and algorithm are rooted on invariant models, providing a new perspective and methodology for OOD learning on graphs. Third, we design and conduct comprehensive experiments on diverse real-world datasets that can reflect in-the-wild nature in real situations (e.g., cross-graph transfers and dynamic evolution) and demonstrate the power of our approach in three scenarios.

## C  PROOFS FOR SECTION 3.1

### C.1  PROOF FOR PROPOSITION 1

We define the aggregated node feature $a_v = \frac{1}{|N_v|} \sum_{u \in N_v} x_u$. According to the definition and setup in Section 3.1, we derive the risk under a specific environment $R(e)$:

$$
\begin{aligned}
&\frac{1}{|V|} \sum_{v \in V} \mathbb{E}_{\mathbf{y}|\mathbf{G_v}=G_v} \left[ \|\hat{y}_v - y_v\|_2^2 \right] \\
=&\frac{1}{|V|} \sum_{v \in V} \mathbb{E}_{\mathbf{n}^1, \mathbf{n}^2} \left[ \|\theta_1 a_v^1 + \theta_2 a_v^2 - y_v\|_2^2 \right] \\
=&\frac{1}{|V|} \sum_{v \in V} \mathbb{E}_{\mathbf{n}^1, \mathbf{n}^2} \left[ \|(\theta_1 + \theta_2 - 1)a_v^1 + \theta_2(n_v^1 + n_v^2 + \epsilon) - n_v^1\|_2^2 \right].
\end{aligned}
\tag{7}
$$

Denote the objective for empirical risk minimization as $L_1 = \mathbb{E}_{\mathbf{e}}[R(e)]$ and we have its first-order derivative w.r.t. $\theta_1$ as

$$
\begin{aligned}
\frac{\partial L_1}{\partial \theta_1} &= \mathbb{E}_{\mathbf{e}}\left[\frac{1}{|V|}\sum_{v\in V}\mathbb{E}_{\mathbf{n}^1,\mathbf{n}^2}\left[2[(\theta_1+\theta_2-1)a_v^1+\theta_2(n_v^1+n_v^2+\epsilon)-n_v^1]\cdot a_v^1\right]\right] \\
&= \mathbb{E}_{\mathbf{e}}\left[\frac{1}{|V|}\sum_{v\in V}\mathbb{E}_{\mathbf{n}^1,\mathbf{n}^2}\left[2[(\theta_1+\theta_2-1)\cdot a_v^1\cdot a_v^1]\right]\right],
\end{aligned}
\tag{8}
$$

where the second step is given by independence among $a_v^1$, $n_v^1$, $n_v^2$ and $\epsilon$. Let $\frac{\partial L_1}{\partial \theta_1}=0$, and we will obtain $\theta_1+\theta_2=1$.

Also, the first-order derivative w.r.t. $\theta_2$ is

$$
\begin{aligned}
\frac{\partial L_1}{\partial \theta_2} &= \mathbb{E}_{\mathbf{e}}\left[\frac{1}{|V|}\sum_{v\in V}\mathbb{E}_{\mathbf{n}^1,\mathbf{n}^2}\left[2[(\theta_1+\theta_2-1)a_v^1+\theta_2(n_v^1+n_v^2+\epsilon)-n_v^1]\cdot(a_v^1+n_v^1+n_v^2+\epsilon)]\right]\right] \\
&= \mathbb{E}_{\mathbf{e}}\left[\frac{1}{|V|}\sum_{v\in V}\mathbb{E}_{\mathbf{n}^1,\mathbf{n}^2}\left[2[(\theta_1+\theta_2-1)\cdot a_v^1\cdot a_v^1+\theta_2(n_v^1\cdot n_v^1+n_v^2\cdot n_v^2+\epsilon\cdot\epsilon)-n_v^1\cdot n_v^1]\right]\right] \\
&= \mathbb{E}_{\mathbf{e}}\left[\frac{1}{|V|}\sum_{v\in V}\mathbb{E}_{\mathbf{n}^1,\mathbf{n}^2}\left[2[(\theta_1+\theta_2-1)\cdot a_v^1\cdot a_v^1+\theta_2(1+1+\sigma_e^2)-1]\right]\right],
\end{aligned}
\tag{9}
$$

where the last step is according to $\mathbb{E}_{\mathbf{x}}[x^2]=\mathbb{E}_{\mathbf{x}}^2[x]+\mathbb{V}_{\mathbf{x}}[x]$. We further let $\frac{\partial L_1}{\partial \theta_2}=0$, and will get the unique solution

$$
\theta_1 = \frac{1+\sigma_e^2}{2+\sigma_e^2}, \quad \theta_2 = \frac{1}{2+\sigma_e^2}.
\tag{10}
$$

## C.2 PROOF FOR PROPOSITION 2

Let $L_2 = \mathbb{V}_{\mathbf{e}}[R(e)] = \mathbb{E}_{\mathbf{e}}[R^2(e)] - \mathbb{E}_{\mathbf{e}}^2[R(e)]$ and $l(e) = (\theta_1+\theta_2-1)a_v^1+\theta_2(n_v^1+n_v^2+\epsilon)-n_v^1$. We derive the first-order derivation of $L_2$ w.r.t. $\theta_1$ and $\theta_2$. Firstly,

$$
\frac{\partial L_2}{\partial \theta_1} = \mathbb{E}_{\mathbf{e}}\left[\frac{1}{|V|}\sum_{v\in V}\mathbb{E}_{\mathbf{n}^1,\mathbf{n}^2}\left[4l^3(e)a_v^1\right]\right] + \mathbb{E}_{\mathbf{e}}^2\left[\frac{1}{|V|}\sum_{v\in V}\mathbb{E}_{\mathbf{n}^1,\mathbf{n}^2}\left[2l(e)a_v^1\right]\right],
\tag{11}
$$

$$
\begin{aligned}
\frac{\partial L_2}{\partial \theta_2} &= \mathbb{E}_{\mathbf{e}}\left[\frac{1}{|V|}\sum_{v\in V}\mathbb{E}_{\mathbf{n}^1,\mathbf{n}^2}\left[4l^3(e)\cdot(a_v^1+n_v^1+n_v^2+\epsilon)\right]\right] \\
&\quad + \mathbb{E}_{\mathbf{e}}^2\left[\frac{1}{|V|}\sum_{v\in V}\mathbb{E}_{\mathbf{n}^1,\mathbf{n}^2}\left[2l(e)\cdot(a_v^1+n_v^1+n_v^2+\epsilon)\right]\right].
\end{aligned}
\tag{12}
$$

By letting $\frac{\partial L_2}{\partial \theta_1}=0$, we obtain the equation $\theta_1+\theta_2=1$. Plugging it into $\frac{\partial L_2}{\partial \theta_2}$ we have

$$
\begin{aligned}
\frac{\partial L_2}{\partial \theta_2} &= \mathbb{E}_{\mathbf{e}}\left[\frac{1}{|V|}\sum_{v\in V}\mathbb{E}_{\mathbf{n}^1,\mathbf{n}^2}\left[4(\theta_2(n_v^1+n_v^2+\epsilon)-n_v^1)^3\cdot(a_v^1+n_v^1+n_v^2+\epsilon)\right]\right] \\
&\quad + \mathbb{E}_{\mathbf{e}}^2\left[\frac{1}{|V|}\sum_{v\in V}\mathbb{E}_{\mathbf{n}^1,\mathbf{n}^2}\left[2(\theta_2(n_v^1+n_v^2+\epsilon)-n_v^1)\cdot(a_v^1+n_v^1+n_v^2+\epsilon)\right]\right],
\end{aligned}
\tag{13}
$$

which is a function of $e$ unless $[\theta_1,\theta_2]=[1,0]$ that gives rise to $\frac{\partial L_2}{\partial \theta_2}=0$ for arbitrary distributions of environments. We thus conclude the proof.

## D PROOFS FOR SECTION 4

### D.1 PROOF FOR THEOREM 1

We first present a useful lemma that interprets the invariance and sufficiency conditions with the terminology of information theory.

**Lemma 1.** *The two conditions in Assumption 1 can be equivalently expressed as 1) (Invariance): $I(\mathbf{y}; \mathbf{e}|\mathbf{r}) = 0$ and 2) (Sufficiency): $I(\mathbf{y}; \mathbf{r})$ is maximized.*

*Proof.* For the invariance, we can easily arrive at the equivalence given the fact

$$I(\mathbf{y}; \mathbf{e}|\mathbf{r}) = D_{KL}(p(\mathbf{y}|\mathbf{e}, \mathbf{r}) \| p(\mathbf{y}|\mathbf{r})). \tag{14}$$

For the sufficiency, we first prove that for $(\mathbf{G_v}, \mathbf{r}, \mathbf{y})$ satisfying that $\mathbf{y} = c^*(\mathbf{r}) + \mathbf{n}$ would also satisfy that $\mathbf{r} = \arg\max_\mathbf{r} I(\mathbf{y}; \mathbf{r})$. We prove it by contradiction. Suppose that $\mathbf{r} \neq \arg\max_\mathbf{r} I(\mathbf{y}; \mathbf{r})$ and $\mathbf{r}' = \arg\max_\mathbf{r} I(\mathbf{y}; \mathbf{r})$ where $r' \neq r$. Then there exists a random variable $\bar{\mathbf{r}}$ such that $\mathbf{r}' = m(\mathbf{r}, \bar{\mathbf{r}})$ where $m$ is a mapping function. We thus have $I(\mathbf{y}; \mathbf{r}') = I(\mathbf{y}; \mathbf{r}, \bar{\mathbf{r}}) = I(c^*(\mathbf{r}); \mathbf{r}, \bar{\mathbf{r}}) = I(c^*(\mathbf{r}); \mathbf{r}) = I(\mathbf{y}; \mathbf{r})$ which leads to contradiction.

We next prove that for $(\mathbf{G_v}, \mathbf{r}, \mathbf{y})$ satisfying that $\mathbf{r} = \arg\max_\mathbf{r} I(\mathbf{y}; \mathbf{r})$ would also satisfy that $\mathbf{y} = c^*(\mathbf{r}) + \mathbf{n}$. Suppose that $\mathbf{y} \neq c^*(\mathbf{r}) + \mathbf{n}$ and $\mathbf{y} = c^*(\mathbf{r}') + \mathbf{n}$ where $\mathbf{r}' \neq \mathbf{r}$. We then have the relationship $I(c^*(\mathbf{r}'); \mathbf{r}) \leq I(c^*(\mathbf{r}'); \mathbf{r}')$ which yields that $\mathbf{r}' = \arg\max_\mathbf{r} I(\mathbf{y}; \mathbf{r})$ and leads to contradiction. □

Given the dependency relationship $\mathbf{z} \leftarrow \mathbf{G_v} \rightarrow \mathbf{y}$, we have the fact that $\max_{q(\mathbf{z}|\mathbf{G_v})} I(\mathbf{y}, \mathbf{z})$ is equivalent to $\min_{q(\mathbf{z}|\mathbf{G_v})} I(\mathbf{y}, \mathbf{G_v}|\mathbf{z})$. Also, we have (treating $q(\mathbf{y}|\mathbf{z})$ as a variational distribution)

$$\begin{aligned} I(\mathbf{y}, \mathbf{G_v}|\mathbf{z}) &= D_{KL}(p(\mathbf{y}|\mathbf{G_v}, \mathbf{e}) \| p(\mathbf{y}|\mathbf{z}, \mathbf{e})) \\ &= D_{KL}(p(\mathbf{y}|\mathbf{G_v}, \mathbf{e}) \| q(\mathbf{y}|\mathbf{z})) - D_{KL}(p(\mathbf{y}|\mathbf{z}, \mathbf{e}) \| q(\mathbf{y}|\mathbf{z})) \\ &\leq D_{KL}(p(\mathbf{y}|\mathbf{G_v}, \mathbf{e}) \| q(\mathbf{y}|\mathbf{z})). \end{aligned} \tag{15}$$

Based on this, we have the inequality

$$I(\mathbf{y}, \mathbf{G_v}|\mathbf{z}) \leq \min_{q(\mathbf{y}|\mathbf{z})} D_{KL}(p(\mathbf{y}|\mathbf{G_v}, \mathbf{e}) \| q(\mathbf{y}|\mathbf{z})). \tag{16}$$

Also, we have (according to our definition in Eq. 6)

$$\begin{aligned} &D_{KL}(p(\mathbf{y}|\mathbf{G_v}, \mathbf{e}) \| q(\mathbf{y}|\mathbf{z})) \\ =&\mathbb{E}_\mathbf{e} \mathbb{E}_{G \sim p_e(\mathbf{G})} \left[ \frac{1}{V} \sum_{v \in V} \mathbb{E}_{y_v \sim p_e(\mathbf{y}|\mathbf{G_v}=G_v)} \mathbb{E}_{z_v \sim q(\mathbf{z}|\mathbf{G_v}=G_v)} \left[ \log \frac{p_e(\mathbf{y} = y_v|\mathbf{G_v} = G_v)}{q(\mathbf{y} = y_v|\mathbf{z} = z_v)} \right] \right] \\ \leq&\mathbb{E}_\mathbf{e} \mathbb{E}_{G \sim p_e(\mathbf{G})} \left[ \frac{1}{V} \sum_{v \in V} \mathbb{E}_{y_v \sim p_e(\mathbf{y}|\mathbf{G_v}=G_v)} \left[ \log \frac{p_e(\mathbf{y} = y_v|\mathbf{G_v} = G_v)}{\mathbb{E}_{z_v \sim q(\mathbf{z}|\mathbf{G_v}=G_v)}[q(\mathbf{y} = y_v|\mathbf{z} = z_v)]} \right] \right], \end{aligned} \tag{17}$$

where the second step is according to Jensen Inequality and the equality holds if $q(\mathbf{z}|\mathbf{G_v})$ is a delta distribution (induced by the GNN encoder $h$). Then the problem $\min_{q(\mathbf{z}|\mathbf{G_v}), q(\mathbf{y}|\mathbf{z})} D_{KL}(p(\mathbf{y}|\mathbf{G_v}, \mathbf{e}) \| q(\mathbf{y}|\mathbf{z}))$ can be equivalently converted into

$$\min_f \mathbb{E}_\mathbf{e} \left[ \frac{1}{|V_e|} \sum_{v \in V_e} l(f(G_v^e), y_v^e) \right] = \min_f \mathbb{E}_\mathbf{e}[L(G^e, Y^e; f)]. \tag{18}$$

We thus have proven that minimizing the expectation term in Eq. 4 is to minimize the upper bound of $I(\mathbf{y}, \mathbf{G_v}|\mathbf{z})$ and contributes to $\max_{q(\mathbf{z}|\mathbf{G_v})} I(\mathbf{y}, \mathbf{z})$.

Second, we have

$$\begin{aligned} &I(\mathbf{y}; \mathbf{e}|\mathbf{z}) \\ =&D_{KL}(p(\mathbf{y}|\mathbf{z}, \mathbf{e}) \| p(\mathbf{y}|\mathbf{z})) \\ =&D_{KL}(p(\mathbf{y}|\mathbf{z}, \mathbf{e}) \| \mathbb{E}_\mathbf{e}[p(\mathbf{y}|\mathbf{z}, \mathbf{e})]) \\ =&D_{KL}(q(\mathbf{y}|\mathbf{z}) \| \mathbb{E}_\mathbf{e}[q(\mathbf{y}|\mathbf{z})]) - D_{KL}(q(\mathbf{y}|\mathbf{z}) \| p(\mathbf{y}|\mathbf{z}, \mathbf{e})) - D_{KL}(\mathbb{E}_\mathbf{e}[p(\mathbf{y}|\mathbf{z}, \mathbf{e})] \| \mathbb{E}_\mathbf{e}[q(\mathbf{y}|\mathbf{z})]) \\ \leq&D_{KL}(q(\mathbf{y}|\mathbf{z}) \| \mathbb{E}_\mathbf{e}[q(\mathbf{y}|\mathbf{z})]). \end{aligned} \tag{19}$$

Besides, we have (according to the definition in Eq. 6)

$$\begin{aligned} &D_{KL}(q(\mathbf{y}|\mathbf{z}) \| \mathbb{E}_\mathbf{e}[q(\mathbf{y}|\mathbf{z})]) \\ =&\mathbb{E}_\mathbf{e} \mathbb{E}_{G \sim p_e(\mathbf{G})} \left[ \frac{1}{|V|} \sum_{v \in V} \mathbb{E}_{y_v \sim p_e(\mathbf{y}|\mathbf{G_v}=G_v)} \mathbb{E}_{z_v \sim q(\mathbf{z}|\mathbf{G_v}=G_v)} \left[ \log \frac{q(\mathbf{y} = y_v|\mathbf{z} = z_v)}{\mathbb{E}_\mathbf{e}[q(\mathbf{y} = y_v|\mathbf{z} = z_v)]} \right] \right]. \end{aligned} \tag{20}$$

Using Jensen Inequality, we will obtain that $D_{KL}(q(\mathbf{y}|\mathbf{z})\|\mathbb{E}_{\mathbf{e}}[q(\mathbf{y}|\mathbf{z})])$ is upper bounded by

$$\mathbb{E}_{\mathbf{e}}[|L(G^e, Y^e; f) - \mathbb{E}_{\mathbf{e}}[L(G^e, Y^e; f)]|] = \mathbb{V}_{\mathbf{e}}[L(G^e, Y^e; f)]. \tag{21}$$

Hence we have proven that minimizing the variance term in Eq. 4 plays a role for solving $\min_{q(\mathbf{z}|\mathbf{G_v})} I(\mathbf{y}; \mathbf{e}|\mathbf{z})$.

### D.2 PROOFS FOR THEOREM 2

With Lemma 1, we know that 1) the representation $\mathbf{z}$ (given by GNN encoder $\mathbf{z} = h(\mathbf{G_v})$) satisfies the invariant condition, i.e., $p(\mathbf{y}|\mathbf{z}) = p(\mathbf{y}|\mathbf{z}, \mathbf{e})$ if and only if $I(\mathbf{y}; \mathbf{e}|\mathbf{z}) = 0$ and 2) the representation $\mathbf{z}$ satisfies the sufficiency condition, i.e., $\mathbf{y} = c^*(\mathbf{z}) + \mathbf{n}$ if and only if $\mathbf{z} = \arg\max_{\mathbf{z}} I(\mathbf{y}; \mathbf{z})$.

We denote the GNN encoder that satisfies the invariance and sufficiency conditions as $h^*$ and the corresponding predictor model $f^*(\mathbf{G_v}) = \mathbb{E}_{\mathbf{y}}[\mathbf{y}|h^*(\mathbf{G_v})]$ with $f^* = c^* \circ h^*$. Since we assume the GNN encoder $q(\mathbf{z}|\mathbf{G_v})$ satisfies the conditions in Assumption 1, then according to Assumption 2, we know that there exists random variable $\overline{\mathbf{z}}$ such that $\mathbf{G_v} = m(\mathbf{z}, \overline{\mathbf{z}})$ and $p(\mathbf{y}|\overline{\mathbf{z}}, \mathbf{e})$ would change arbitrarily across environments. Based on this, for any environment $e$ that gives the distribution $p_e(\mathbf{y}, \mathbf{z}, \overline{\mathbf{z}})$, we can construct environment $e'$ with the distribution $p_{e'}(\mathbf{y}, \mathbf{z}, \overline{\mathbf{z}})$ that satisfies

$$p_{e'}(\mathbf{y}, \mathbf{z}, \overline{\mathbf{z}}) = p_e(\mathbf{y}, \mathbf{z})p_{e'}(\overline{\mathbf{z}}). \tag{22}$$

Then we follow the reasoning line of Theorem 2.1 in Liu et al. (2021) to finish the proof by showing that for arbitrary function $f = c \circ h$ and environment $e$, there exists an environment $e'$ such that

$$\mathbb{E}_{G \sim p_{e'}(\mathbf{G})}\left[\frac{1}{|V|}\sum_{v \in V}\mathbb{E}_{y_v \sim p_{e'}(\mathbf{y}|\mathbf{G_v}=G_v)}[l(f(G_v), y_v)]\right]$$
$$\geq \mathbb{E}_{G \sim p_e(\mathbf{G})}\left[\frac{1}{|V|}\sum_{v \in V}\mathbb{E}_{y \sim p_e(\mathbf{y}|\mathbf{G_v}=G_v)}[l(f^*(G_v), y_v)]\right]. \tag{23}$$

Concretely we have

$$\mathbb{E}_{G \sim p_{e'}(\mathbf{G})}\left[\frac{1}{|V|}\sum_{v \in V}\mathbb{E}_{(y_v, z_v, \overline{z}_v) \sim p_{e'}(\mathbf{y}, \mathbf{z}, \overline{\mathbf{z}}|\mathbf{G_v}=G_v)}[l(c(z_v, \overline{z}_v), y_v)]\right]$$

$$= \mathbb{E}_{G' \sim p_{e'}(\mathbf{G})}\left[\frac{1}{|V'|}\sum_{v' \in V'}\mathbb{E}_{\overline{z}_{v'} \sim p_{e'}(\overline{\mathbf{z}}|\mathbf{G_v}=G_{v'})}\mathbb{E}_{G \sim p_e(\mathbf{G})}\left[\frac{1}{|V|}\sum_{v \in V}\mathbb{E}_{(y_v, z_v) \sim p_e(\mathbf{y}, \mathbf{z}|\mathbf{G_v}=G_v)}[l(c(z_v, \overline{z}_{v'}), y_v)]\right]\right]$$

$$\geq \mathbb{E}_{G' \sim p_{e'}(\mathbf{G})}\left[\frac{1}{|V'|}\sum_{v' \in V'}\mathbb{E}_{\overline{z}_{v'} \sim p_{e'}(\overline{\mathbf{z}}|\mathbf{G_v}=G_{v'})}\mathbb{E}_{G \sim p_e(\mathbf{G})}\left[\frac{1}{|V|}\sum_{v \in V}\mathbb{E}_{(y_v, z_v) \sim p_e(\mathbf{y}, \mathbf{z}|\mathbf{G_v}=G_v)}[l(c^*(z_v, \overline{z}_{v'}), y_v)]\right]\right]$$

$$= \mathbb{E}_{G' \sim p_{e'}(\mathbf{G})}\left[\frac{1}{|V'|}\sum_{v' \in V'}\mathbb{E}_{\overline{z}_{v'} \sim p_{e'}(\overline{\mathbf{z}}|\mathbf{G_v}=G_{v'})}\mathbb{E}_{G \sim p_e(\mathbf{G})}\left[\frac{1}{|V|}\sum_{v \in V}\mathbb{E}_{(y_v, z_v) \sim p_e(\mathbf{y}, \mathbf{z}|\mathbf{G_v}=G_v)}[l(c^*(z_v), y_v)]\right]\right]$$

$$= \mathbb{E}_{G \sim p_e(\mathbf{G})}\left[\frac{1}{|V|}\sum_{v \in V}\mathbb{E}_{(y_v, z_v) \sim p_e(\mathbf{y}, \mathbf{z}|\mathbf{G_v}=G_v)}[l(c^*(z_v), y_v)]\right]$$

$$= \mathbb{E}_{G \sim p_e(\mathbf{G})}\left[\frac{1}{|V|}\sum_{v \in V}\mathbb{E}_{(y_v, z_v, \overline{z}_v) \sim p_e(\mathbf{y}, \mathbf{z}, \overline{\mathbf{z}}|\mathbf{G_v}=G_v)}[l(c^*(z_v), y_v)]\right], \tag{24}$$

where the first equality is given by Eq. 22 and the second/third steps are due to the sufficiency condition of $h^*$.

### D.3 PROOF FOR THEOREM 3

Recall that according to our definition in Eq. 6, the KL divergence $D_{KL}(p_e(\mathbf{y}|\mathbf{G_v})\|q(\mathbf{y}|\mathbf{G_v}))$ would be

$$D_{KL}(p_e(\mathbf{y}|\mathbf{G_v})\|q(\mathbf{y}|\mathbf{G_v})) := \mathbb{E}_{G\sim p_e(\mathbf{G})}\left[\frac{1}{|V|}\sum_{v\in V}\mathbb{E}_{y_v\sim p_e(\mathbf{y}|\mathbf{G_v}=G_v)}\left[\log\frac{p_e(\mathbf{y}=y_v|\mathbf{G_v}=G_v)}{q(\mathbf{y}=y_v|\mathbf{G_v}=G_v)}\right]\right].\tag{25}$$

Based on this newly defined KL divergence, we can extend the information-theoretic framework (Federici et al., 2021) for analysis on graph data. First, we can decompose the training error (resp. OOD error) into a representation error and a latent predictive error.

**Lemma 2.** *For any GNN encoder $q(\mathbf{z}|\mathbf{G_v})$ and classifier $q(\mathbf{y}|\mathbf{z})$, we have*

$$D_{KL}(p_e(\mathbf{y}|\mathbf{G_v})\|q(\mathbf{y}|\mathbf{G_v})) \leq I_e(\mathbf{G_v};\mathbf{y}|\mathbf{z}) + D_{KL}(p_e(\mathbf{y}|\mathbf{z})\|q(\mathbf{y}|\mathbf{z})),\tag{26}$$

$$D_{KL}(p_{e'}(\mathbf{y}|\mathbf{G_v})\|q(\mathbf{y}|\mathbf{G_v})) \leq I_{e'}(\mathbf{G_v};\mathbf{y}|\mathbf{z}) + D_{KL}(p_{e'}(\mathbf{y}|\mathbf{z})\|q(\mathbf{y}|\mathbf{z})).\tag{27}$$

*Proof.* Firstly, we have

$$
\begin{aligned}
&D_{KL}(p_e(\mathbf{y}|\mathbf{G_v})\|q(\mathbf{y}|\mathbf{G_v}))\\
=&\mathbb{E}_{G\sim p_e(\mathbf{G})}\left[\frac{1}{|V|}\sum_{v\in V}\mathbb{E}_{y_v\sim p_e(\mathbf{y}|\mathbf{G_v}=G_v)}\left[\log\frac{p_e(\mathbf{y}=y_v|\mathbf{G_v}=G_v)}{q(\mathbf{y}=y_v|\mathbf{G_v}=G_v)}\right]\right]\\
=&\mathbb{E}_{G\sim p_e(\mathbf{G})}\left[\frac{1}{|V|}\sum_{v\in V}\mathbb{E}_{y_v\sim p_e(\mathbf{y}|\mathbf{G_v}=G_v)}\left[\log\frac{p_e(\mathbf{y}=y_v|\mathbf{G_v}=G_v)}{\mathbb{E}_{z_v\sim q(\mathbf{z}|\mathbf{G_v}=G_v)}q(\mathbf{y}=y_v|\mathbf{z}=z_v)}\right]\right]\\
\leq&\mathbb{E}_{G\sim p_e(\mathbf{G})}\left[\frac{1}{|V|}\sum_{v\in V}\mathbb{E}_{y_v\sim p_e(\mathbf{y}|\mathbf{G_v}=G_v)}\mathbb{E}_{z_v\sim q(\mathbf{z}|\mathbf{G_v}=G_v)}\left[\log\frac{p_e(\mathbf{y}=y_v|\mathbf{G_v}=G_v)}{q(\mathbf{y}=y_v|\mathbf{z}=z_v)}\right]\right]\\
=&D_{KL}(p_e(\mathbf{y}|\mathbf{G_v})\|q(\mathbf{y}|\mathbf{z})),
\end{aligned}\tag{28}
$$

where the third step is again due to Jensen Inequality and the equality holds once $q(\mathbf{z}|\mathbf{G_v})$ is a delta distribution.

Besides, we have

$$
\begin{aligned}
&D_{KL}(p_e(\mathbf{y}|\mathbf{G_v})\|q(\mathbf{y}|\mathbf{z}))\\
=&\mathbb{E}_{G\sim p_e(\mathbf{G})}\left[\frac{1}{|V|}\sum_{v\in V}\mathbb{E}_{y_v\sim p_e(\mathbf{y}|\mathbf{G_v}=G_v)}\mathbb{E}_{z_v\sim q(\mathbf{z}|\mathbf{G_v}=G_v)}\left[\log\frac{p_e(\mathbf{y}=y_v|\mathbf{G_v}=G_v)}{q(\mathbf{y}=y_v|\mathbf{z}=z_v)}\right]\right]\\
=&\mathbb{E}_{G\sim p_e(\mathbf{G})}\left[\frac{1}{|V|}\sum_{v\in V}\mathbb{E}_{y_v\sim p_e(\mathbf{y}|\mathbf{G_v}=G_v)}\mathbb{E}_{z_v\sim q(\mathbf{z}|\mathbf{G_v}=G_v)}\left[\log\frac{p_e(\mathbf{y}=y_v|\mathbf{G_v}=G_v)p_e(\mathbf{y}=y_v|\mathbf{z}=z_v)}{p_e(\mathbf{y}=y_v|\mathbf{z}=z_v)q(\mathbf{y}=y_v|\mathbf{z}=z_v)}\right]\right]\\
=&I(\mathbf{G_v};\mathbf{y}|\mathbf{z}) + D_{KL}(p_e(\mathbf{y}|\mathbf{z})\|q(\mathbf{y}|\mathbf{z})).
\end{aligned}\tag{29}
$$

The result for $D_{KL}(p_{e'}(\mathbf{y}|\mathbf{G_v})\|q(\mathbf{y}|\mathbf{G_v}))$ can be obtained in a similar way. □

**Lemma 3.** *For any $q(\mathbf{z}|\mathbf{G_v})$ and $q(\mathbf{y}|\mathbf{z})$, the following inequality holds for any $z$ satisfying $p(\mathbf{z} = z|\mathbf{e} = e) > 0$, $\forall e \in \mathcal{E}$.*

$$D_{JSD}(p_{e'}(\mathbf{y}|\mathbf{z})\|q(\mathbf{y}|\mathbf{z})) \leq \left(\sqrt{\frac{1}{2\alpha}I(\mathbf{y};\mathbf{e}|\mathbf{z})} + \sqrt{\frac{1}{2}D_{KL}(p_e(\mathbf{y}|\mathbf{z})\|q(\mathbf{y}|\mathbf{z}))}\right)^2.\tag{30}$$

*Proof.* The proof can be adapted by from the Proposition 3 in Federici et al. (2021) by replacing $\mathbf{e}$ in our case with $\mathbf{t}$.

□

Table 4: Statistic information for `Twitch-Explicit` datasets.

| Dataset | #Nodes | #Edges | #Density | Avg Degree | Max Degree |
|---------|--------|--------|----------|------------|------------|
| DE | 9498 | 153138 | 0.0033 | 16 | 3475 |
| ENGB | 7126 | 35324 | 0.0013 | 4 | 465 |
| ES | 4648 | 59382 | 0.0054 | 12 | 809 |
| FR | 6549 | 112666 | 0.0052 | 17 | 1517 |
| PTBR | 1912 | 31299 | 0.0171 | 16 | 455 |
| RU | 4385 | 37304 | 0.0038 | 8 | 575 |
| TW | 2772 | 63462 | 0.0165 | 22 | 1171 |

The results of Lemma 2 and 3 indicate that if we aim to reduce the OOD error measured by $D_{KL}(p_{e'}(\mathbf{y}|\mathbf{G_v})\|q(\mathbf{y}|\mathbf{G_v}))$, one need to control three terms: 1) $I_e(\mathbf{G_v};\mathbf{y}|\mathbf{z})$, 2) $D_{KL}(p_e(\mathbf{y}|\mathbf{z})\|q(\mathbf{y}|\mathbf{z})$ and 3) $I(\mathbf{y};\mathbf{e}|\mathbf{z})$. The next lemma unifies minimization for the first two terms.

**Lemma 4.** *For any $q(\mathbf{z}|\mathbf{G_v})$ and $q(\mathbf{y}|\mathbf{z})$, we have*

$$\min_{q(\mathbf{z}|\mathbf{G_v}),q(\mathbf{y}|\mathbf{z})} D_{KL}(p_e(\mathbf{y}|\mathbf{G_v})\|q(\mathbf{y}|\mathbf{z})) \Leftrightarrow \min_{q(\mathbf{z}|\mathbf{G_v})} I_e(\mathbf{G_v};\mathbf{y}|\mathbf{z}) + \min_{q(\mathbf{y}|\mathbf{z})} D_{KL}(p_e(\mathbf{y}|\mathbf{z})\|q(\mathbf{y}|\mathbf{z})). \tag{31}$$

*Proof.* Recall that $q(\mathbf{y}|\mathbf{z})$ is a variational distribution. We have

$$\begin{aligned} I_e(\mathbf{G_v};\mathbf{y}|\mathbf{z}) &= D_{KL}(p_e(\mathbf{y}|\mathbf{G_v})\|p_e(\mathbf{y}|\mathbf{z}))) \\ &= D_{KL}(p_e(\mathbf{y}|\mathbf{G_v})\|q(\mathbf{y}|\mathbf{z})) - D_{KL}(p_e(\mathbf{y}|\mathbf{z})\|q(\mathbf{y}|\mathbf{z})) \\ &\leq D_{KL}(p_e(\mathbf{y}|\mathbf{G_v})\|q(\mathbf{y}|\mathbf{z})). \end{aligned} \tag{32}$$

Therefore, we can see that $I_e(\mathbf{G_v};\mathbf{y}|\mathbf{z})$ is upper bounded by $D_{KL}(p_e(\mathbf{y}|\mathbf{G_v}\|q(\mathbf{y}|\mathbf{z}))$ and the equality holds if and only if $D_{KL}(p_e(\mathbf{y}|\mathbf{z})\|q(\mathbf{y}|\mathbf{z})) = 0$. We thus conclude the proof. □

Recall that according to Lemma 1 we have the fact that our objective in Eq. 4 essentially has the similar effect as

$$\min_{q(\mathbf{z}|\mathbf{G_v}),q(\mathbf{y}|\mathbf{z})} D_{KL}(p_e(\mathbf{y}|\mathbf{G_v})\|q(\mathbf{y}|\mathbf{z})) + I(\mathbf{y};\mathbf{e}|\mathbf{z}). \tag{33}$$

Based on the Lemma 2, 3 and 4, we know that optimization for the objective Eq. 4 can reduce the upper bound of OOD error given by $D_{KL}(p_{e'}(\mathbf{y}|\mathbf{G_v})\|q(\mathbf{y}|\mathbf{G_v})$ on condition that $I_{e'}(\mathbf{G_v};\mathbf{y}|\mathbf{z}) = I_e(\mathbf{G_v};\mathbf{y}|\mathbf{z})$. We conclude our proof for Theorem 3.

# E  DATASETS AND EVALUATION PROTOCOLS

In this section, we introduce the detailed information for experimental datasets and also provide the details for our evaluation protocols including data preprocessing, dataset splits and the ways for calculating evaluation metrics. In the following subsections, we present the information for the three scenarios, respectively.

## E.1  ARTIFICIAL DISTRIBUTION SHIFTS ON CORA AND AMAZON-PHOTO

`Cora` and `Amazon-Photo` are two commonly used node classification benchmarks and widely adopted for evaluating the performance of GNN designs. These datasets are of medium size with thousands of nodes. See Table 1 for more statistic information. `Cora` is a citation network where nodes represent papers and edges represent their citation relationship. `Amazon-Photo` is a co-purchasing network where nodes represent goods and edges indicate that two goods are frequently bought together. In the original dataset, the available node features have strong correlation with node labels. To evaluate model's ability for out-of-distribution generalization, we need to introduce distribution shifts into the training and testing data.

For each dataset, we use the provided node features to construct node labels and spurious environment-sensitive features. Specifically, assume the provided node features as $X_1$. Then we adopt a randomly

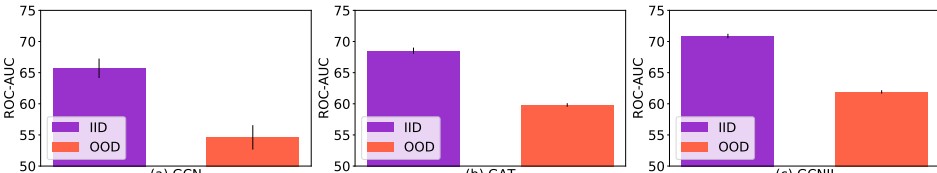

Figure 6: Comparison of different leave-out data on `Twitch-Explicit`. We consider three GNN backbones trained with ERM. The "OOD" means that we train the model on one graph DE and report the metric on another graph ENGB. The "IID" means that we train the model on 90% nodes of DE and report the metric on the remaining nodes. The results clearly show that the model performance suffers a significantly drop from the case "IID" to the case "OOD". This indicates that the graph-level splitting for training/validation/testing splits used in Section 5.2 indeed introduces distribution shifts and would require the model to deal with out-of-distribution data during test.

initialized GNN (with input of $X_1$ and adjacency matrix) to generate node labels $Y$ (via taking an argmax in the output layer to obtain one-hot vectors), and another randomly initialized GNN (with input of the concatenation of $Y$ and an environment id) to generate spurious node features $X_2$. After that, we concatenate two portions of features $X = [X_1, X_2]$ as input node features for training and evaluation. In this way, we construct ten graphs with different environment id's for each dataset. We use one graph for training, one for validation and report the classification accuracy on the remaining graphs. One may realize that this data generation is a generalized version of our motivating example in Section 3.1 and we replace the linear aggregation as a randomly initialized graph neural network to introduce non-linearity.

In fact, with our data generation, the original node features $X_1$ can be seen as domain-invariant features that are sufficiently predictive for node labels and insensitive to different environments, while the generated features $X_2$ are domain-variant features that are conditioned on environments. Therefore, in principle, the ideal case for the model is to identify and leverage the invariant features for prediction. In practice, there exist multiple factors that may affect model's learning, including the local optimum and noise in data. Therefore, one may not expect the model to exactly achieve the ideal case since there also exists useful predictive information in $X_2$ that may help the model to increase the training accuracy. Yet, through our experiments in Fig. 2(b) and 3(b), we show that the reliance of EERM on spurious features is much less than ERM, which we believe could serve as concrete evidence that our approach is capable for guiding the GNN model to alleviate reliance on domain-variant features.

### E.2 CROSS-DOMAIN TRANSFERS ON MULTI-GRAPH DATA

A typical scenario for distribution shifts on graphs is the problem of cross-domain transfers. There are quite a few real-world situations where one has access to multiple observed graphs each of which is from a specific domain. For example, in social networks, the domains can be instantiated as where or when the networks are collected. In protein networks, there may exist observed graph data (protein-protein interactions) from distinct species which can be seen as distinct domains. In short, since most of graph data records the relational structures among a specific group of entities and the interactions/relationships among entities from different groups often have distinct characteristics, the data-generating distributions would vary across groups, which bring up domain shifts.

Yet, to enable transfer learning across graphs, the graphs in one dataset need to share the same input feature space and output space. We adopt two public datasets `Twitch-Explicit` and `Facebook-100` that satisfy this requirement.

`Twitch-Explicit` contains seven networks where nodes represent Twitch users and edges represent their mutual friendships. Each network is collected from a particular region, including DE, ENGB, ES, FR, PTBR, RU and TW. These seven networks have similar sizes and different densities and maximum node degrees, as shown in Table 4. Also, in Fig. 6, we compare the ROC-AUC results on different leave-out data. We consider GCN, GAT and GCNII as the GNN backbones and train the model with standard empirical risk minimization (ERM). We further consider two ways for data splits. In the first case, which we call "OOD", we train the model on the nodes of one graph DE and

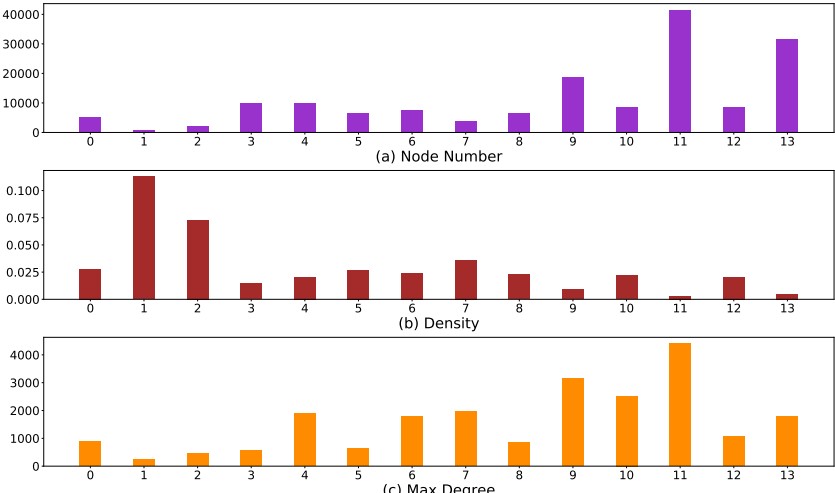

Figure 7: Comparison of node numbers, densities and maximum node degrees of fourteen graphs used in our experiments on `Facebook-100`. The index 0-13 stand for John Hopkins, Caltech, Amherst, Bingham, Duke, Princeton, WashU, Brandeis, Carnegie, Cornell, Yale, Penn, Brown and Texas, respectively. As we can see, these graphs have very distinct statistics, which indicates that there exist distribution shifts w.r.t. graph structures.

report the highest ROC-AUC on the nodes of another graph ENGB. In the second case, which we call "IID", we train the model on 90% nodes of DE and evaluate the performance on the leave-out 10% data. The results in Fig. 6 show that the model performance exhibits a clear drop from "IID" to "OOD", which indicates that there indeed exist distribution shifts among different input graphs. This also serves as a justification for our evaluation protocol in Section 5.2 where we adopt the graph-level splitting to construct training/validation/testing sets.

Another dataset is `Facebook-100` which consists of 100 Facebook friendship network snapshots from the year 2005, and each network contains nodes as Facebook users from a specific American university. We adopt fourteen networks in our experiments: John Hopkins, Caltech, Amherst, Bingham, Duke, Princeton, WashU, Brandeis, Carnegie, Cornell, Yale, Penn, Brown and Texas. Recall that in Section 5.2 we use Penn, Brown and Texas for testing, Cornell and Yale for validation, and use three different combinations from the remaining graphs for training. These graphs have significantly diverse sizes, densities and degree distributions. In Fig. 7 we present a comparison which indicates that the distributions of graph structures among these graphs are different. Concretely, the testing graphs Penn and Texas are much larger (with 41554 and 31560 nodes, respectively) than training/validation graphs (most with thousands of nodes). Also, the training graphs Caltech and Amherst are much denser than other graphs in the dataset, while some graphs like Penn have nodes with very large degrees. These statistics suggest that our evaluation protocol requires the model to handle different graph structures from training/validation to testing data.

### E.3   TEMPORAL EVOLUTION ON DYNAMIC GRAPH DATA

Another common scenario is for temporal graphs that dynamically evolve as time goes by. The types of evolution can be generally divided into two categories. In the first case, there are multiple graph snapshots and each snapshot is taken at one time. As time goes by, there exists a sequence of graph snapshots which may contain different node sets and data distributions. Typical examples include financial networks that record the payment flows among transactions within different time intervals. In the second case, there is one graph that evolves with node/edge adding or deleting. Typical examples include some large-scale real-world graphs like social networks and citation networks where the distribution for node features, edges and labels would have strong correlation with time (in different scales). We adopt two public real-world datasets `Elliptic` and `OGB-Arxiv` for node classification experiments.

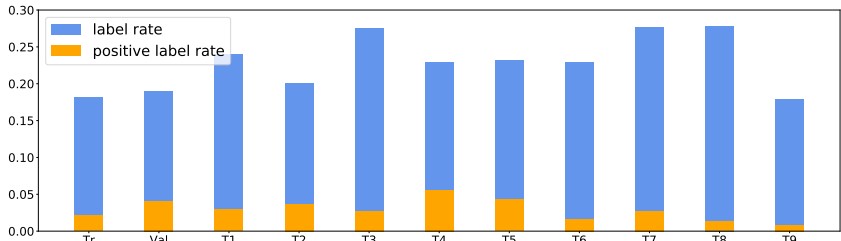

Figure 8: The label rates and positive label rates of training/validation/testing data splits of `Elliptic`. The positive class (illicit transaction) and negative class (licit transaction) are very imbalanced. Also, in different splits, the distributions for labels exhibit clear differences.

`Elliptic` contains a sequence of 49 graph snapshots. Each graph snapshot is a network of Bitcoin transactions where each node represents one transaction and each edge indicates a payment flow. Approximately 20% of the transactions are marked with licit or illicit ones and the goal is to identify illicit transaction in the future observed network. Since in the original dataset, the first six snapshots have extremely imbalanced classes (where the illicit transactions are less than 10 among thousands of nodes), we remove them and use the 7th-11th/12th-17th/17th-49th snapshots for training/validation/testing. Also, due to the fact that each graph snapshot has very low positive label rate, we group the 33 testing graph snapshots into 9 test sets according to the chronological order. In Fig. 8 we present the label rate and positive label rate for training/validation/testing sets. As we can see, the positive label rates are quite different in different data sets. Indeed, the model needs to handle distinct label distributions from training to testing data.

`OGB-Arxiv` is composed of 169,343 Arxiv CS papers from 40 subject areas and their citation relationship. The goal is to predict a paper's subject area. In (Hu et al., 2020), the papers published before 2017, on 2018 and since 2019 are used for training/validation/testing. Also, the authors adopt the transductive learning setting, i.e., the nodes in validation and test sets also exist in the graph for training. In our case, we instead adopt inductive learning setting where the nodes in validation and test sets are unseen during training, which is more akin to the real-world situation. Besides, for better evaluation on generalization, especially extrapolating to new data, we consider dataset splits with a larger year gap: we use papers published before 2011 for training, from 2011 to 2014 for validation, and after 2014 for test. Such a dataset splitting way would introduce distribution shift between training and testing data, since several latent influential factors (e.g., the popularity of research topics) for data generation would change over time. In Fig. 9, we visualize the T-SNE embeddings of the nodes and mark the training/validation/testing nodes with different colors. From Fig. 9(a) to Fig. 9(c), we can see that testing nodes non-overlapped with the training/validation ones exhibit an increase, which suggests that the distribution shifts enlarge as time difference goes large. This phenomenon echoes the results we achieve in Table 3 where we observe that as the time difference between testing and training data goes larger, model performance suffers a clear drop, with ERM suffering more than EERM.

## F  IMPLEMENTATION DETAILS

In this section, we present the details for our implementation in Section 5 including the model architectures, hyper-parameter settings and training details in order for reproducibility. Most of our experiments are run on GeForce RTX 2080Ti with 11GB except some experiments requiring large GPU memory for which we adopt RTX 8000 with 48GB. The configurations of our environments and packages are listed below:

- Ubuntu 16.04
- CUDA 10.2
- PYTHON 3.7
- Numpy 1.20.3
- PyTorch 1.9.0

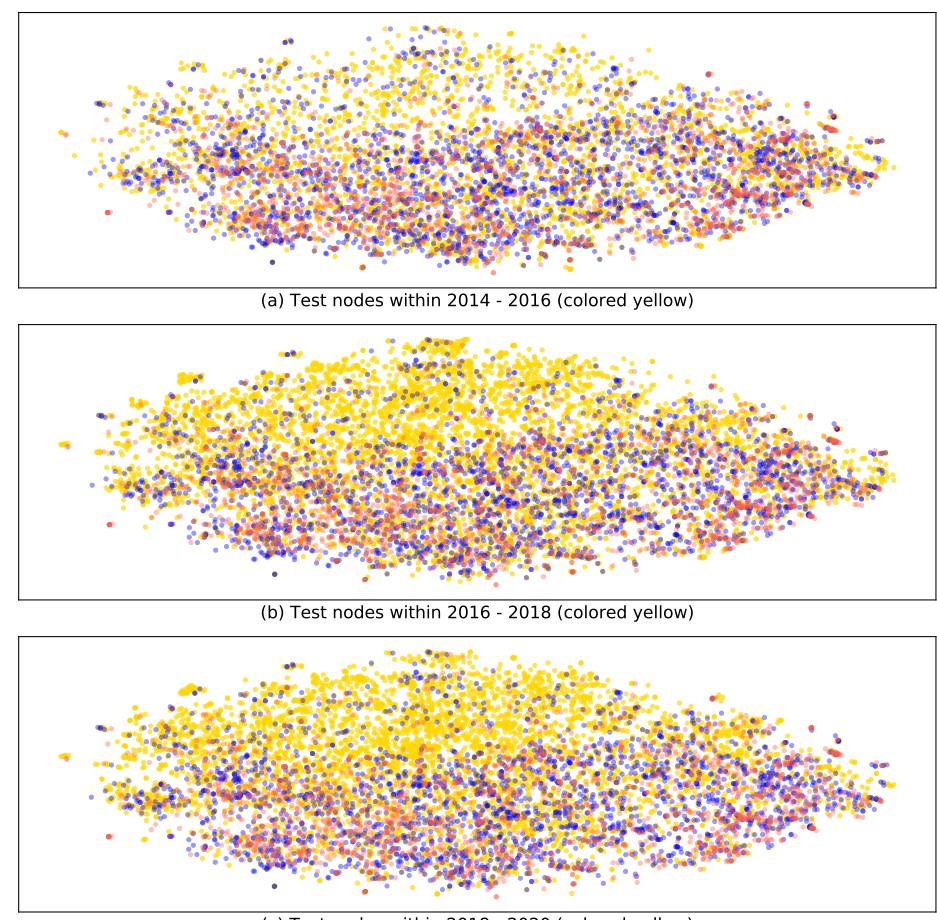

(a) Test nodes within 2014 - 2016 (colored yellow)

(b) Test nodes within 2016 - 2018 (colored yellow)

(c) Test nodes within 2018 - 2020 (colored yellow)

Figure 9: T-SNE visualization of training/validation/testing nodes in `OGB-Arxiv`. We mark training nodes (within 1950-2011) and validation nodes (within 2011-2014) as red and blue, respectively. In (a)-(c), the test nodes within different time intervals are visualized as yellow points. We can see that as the time difference of testing data and training/validation data goes large from (a) to (c), the testing nodes non-overlapped with training/validation ones become more, which suggests that the distribution shifts become more significant and require the model to extrapolate to more difficult future data.

- PyTorch Geometric 1.7.2

### F.1 MODEL ARCHITECTURES

In our experiments in Section 5, we adopt different GNN architectures as the backbone. Here we introduce the details for them.

**GCN.** We use the `GCNConv` available in Pytorch Geometric for implementation. The detailed architecture description is as below:

- A sequence of $L$-layer `GCNConv`.
- Add self-loop and use batch normalization for graph convolution in each layer.
- Use ReLU as the activation.

**GAT.** We use the `GATConv` available in Pytorch Geometric for implementation. The detailed architecture description is as below:

- A sequence of $L$-layer `GATConv` with head number $H$.
- Add self-loop and use batch normalization for graph convolution in each layer.
- Use ELU as the activation.

**GraphSAGE.** We use the `SAGEConv` available in Pytorch Geometric for implementation. The detailed architecture description is as below:

- A sequence of $L$-layer `SAGEConv`.
- Add self-loop and use batch normalization for graph convolution in each layer.
- Use ReLU as the activation.

**GCNII.** We use the implementation[4] provided by the original paper (Chen et al., 2020a). The associated hyper-parameters in GCNII model are set as: $\alpha_{GCNII} = 0.1$ and $\lambda_{GCNII} = 1.0$.

**GPRGNN.** We use the implementation[5] provided by Chien et al. (2021). We adopt the `PPR` initialization and `GPRprop` as the propagation unit. The associated hyper-parameters in GPRGNN model are set as: $\alpha_{GPRGNN} = 0.1$.

### F.2 HYPER PARAMETER SETTINGS

The hyper-parameters for model architectures are set as default values in different cases. Other hyper-parameters are searched with grid search on validation dataset. The searching space are as follows: learning rate for GNN backbone $\alpha_f \in \{0.0001, 0.0002, 0.001, 0.005, 0.01\}$, learning rate for graph editers $\alpha_g \in \{0.0001, 0.001, 0.005, 0.01\}$, weight for combination $\beta \in \{0.2, 0.5, 1.0, 2.0, 3.0\}$, number of edge editing for each node $s \in \{1, 5, 10\}$, number of iterations for inner update before one-step outer update $T \in \{1, 5\}$.

#### F.2.1 SETTINGS FOR SECTION 5.1

We consider 2-layer GCN with hidden size 32. We use weight decay with coefficient set as 1e-3. Besides, we set $\alpha_g = 0.005$, $\alpha_f = 0.01$, $\beta = 2.0$, $s = 5$, $T = 1$.

#### F.2.2 SETTINGS FOR SECTION 5.2

For GCN, we set the layer number $L$ as 2. For GAT, we set $L = 2$ and $H = 4$. For GCNII, we set the layer number as 10. We use hidden size 32 and weight decay with coefficient set as 1e-3.

For `Twitch-Explicit`, other hyper-parameters are set as follows:

- GCN: $\alpha_g = 0.001$, $\alpha_f = 0.01$, $\beta = 3.0$, $s = 5$, $T = 1$.
- GAT: $\alpha_g = 0.005$, $\alpha_f = 0.01$, $\beta = 1.0$, $s = 5$, $T = 1$.
- GCNII: $\alpha_g = 0.01$, $\alpha_f = 0.001$, $\beta = 1.0$, $s = 5$, $T = 1$.

For `Facebook-100`, other hyper-parameters are set as: $\alpha_g = 0.005$, $\alpha_f = 0.01$, $\beta = 1.0$, $s = 5$, $T = 1$.

#### F.2.3 SETTINGS FOR SECTION 5.3

For GraphSAGE and GPRGNN, we set the layer number as 5 and hidden size as 32.

For `Elliptic`, other hyper-parameters are set as follows:

- GraphSAGE: $\alpha_g = 0.0001$, $\alpha_f = 0.0002$, $\beta = 1.0$, $s = 5$, $T = 1$.
- GPRGNN: $\alpha_g = 0.005$, $\alpha_f = 0.01$, $\beta = 1.0$, $s = 5$, $T = 1$.

For `OGB-Arxiv`, other hyper-parameters are set as follows:

---

[4]https://github.com/chennnM/GCNII
[5]https://github.com/jianhao2016/GPRGNN

- GraphSAGE: $\alpha_g = 0.01$, $\alpha_f = 0.005$, $\beta = 0.5$, $s = 1$, $T = 5$.
- GPRGNN: $\alpha_g = 0.001$, $\alpha_f = 0.01$, $\beta = 1.0$, $s = 1$, $T = 5$.

### F.3   TRAINING DETAILS

For each method, we train the model with a fixed number of epochs and report the test result achieved at the epoch when the model provides the best performance on validation set.

## G   MORE EXPERIMENT RESULTS

We provide additional experiment results in this section. In Fig.10 and 11 we present the distribution of test accuracy on Cora when using SGC and GAT, respectively, as the GNNs for data generation. In Fig. 12 and 13 we further compare with the training accuracy using all the features and removing the spurious ones for inference. These results are consistent with those presented in Section 5.1, which again verifies the effectiveness of our approach. Besides, the corresponding extra results on Photo are shown in Fig. 14, 15, 16 and 17, which also back up our discussions in Section 5.1.

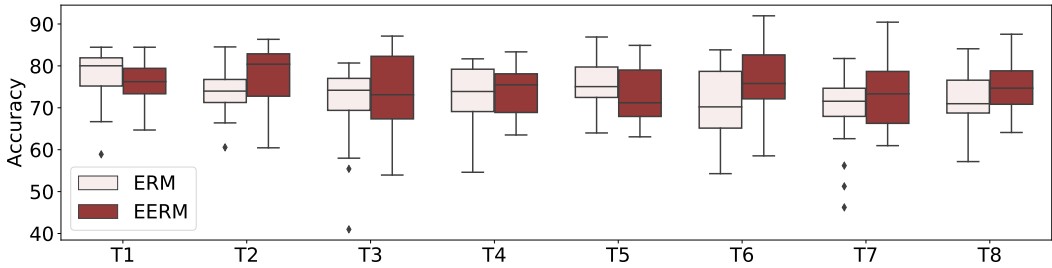

Figure 10: Distribution of test accuracy results on `Cora` with artificial distribution shifts generated by SGC as the GNN generator.

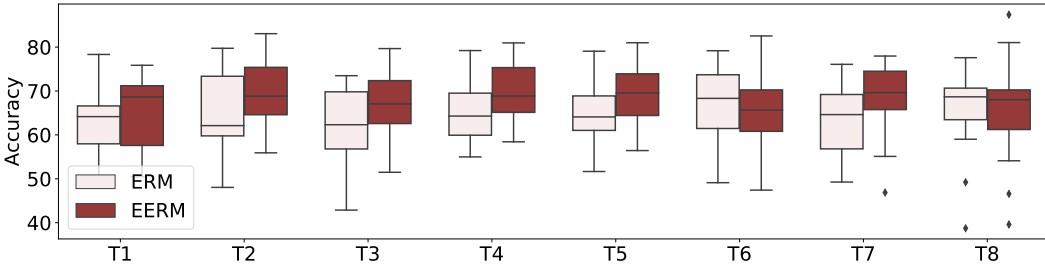

Figure 11: Distribution of test accuracy results on `Cora` with artificial distribution shifts generated by GAT as the GNN generator.

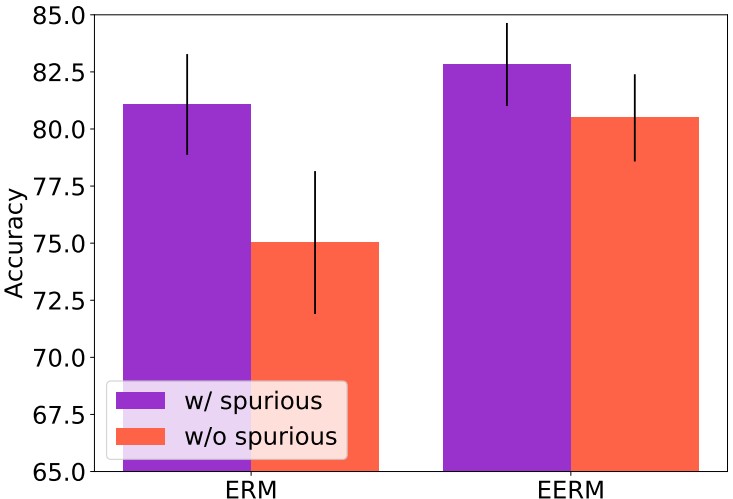

Figure 12: Comparison of training accuracy using all the features v.s. removing the spurious features for inference on `Cora` with artificial distribution shifts generated by SGC as the GNN generator.

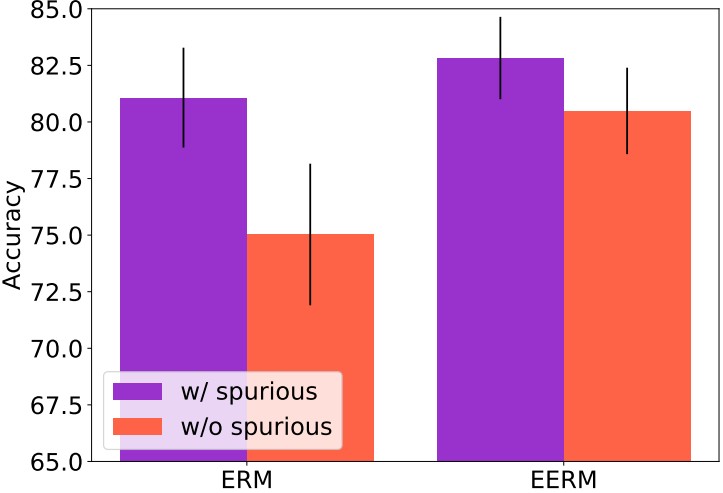

Figure 13: Comparison of training accuracy using all the features v.s. removing the spurious features for inference on `Cora` with artificial distribution shifts generated by GAT as the GNN generator.

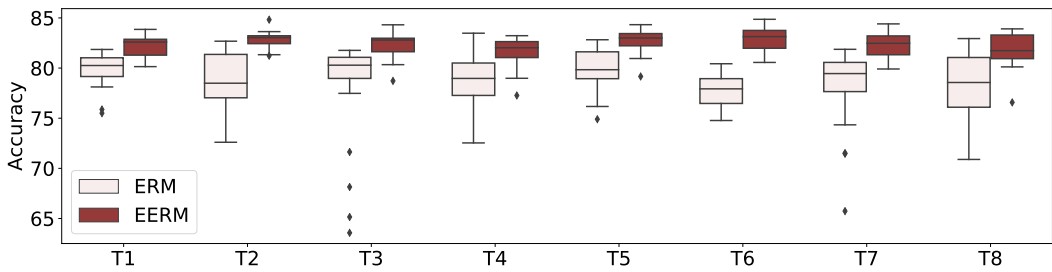

Figure 14: Distribution of test accuracy results on `Photo` with artificial distribution shifts generated by SGC as the GNN generator.

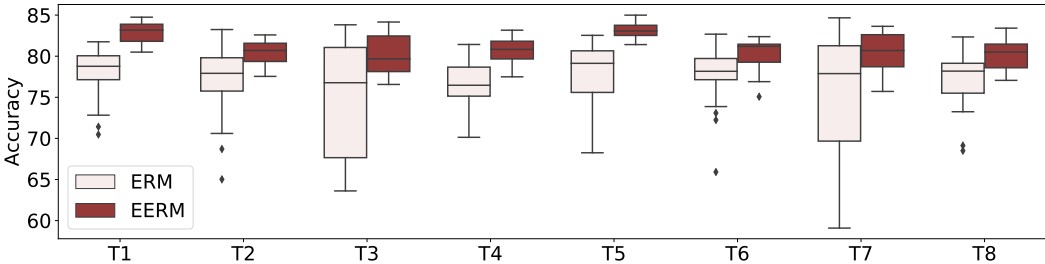

Figure 15: Distribution of test accuracy results on `Photo` with artificial distribution shifts generated by GAT as the GNN generator.

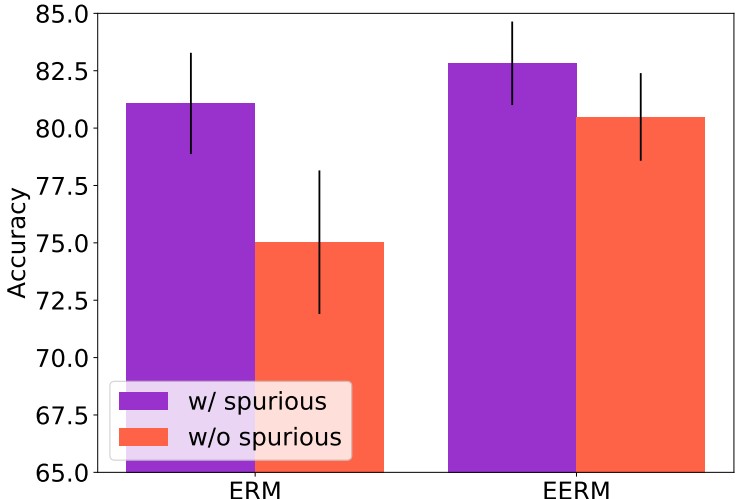

Figure 16: Comparison of training accuracy using all the features v.s. removing the spurious features for inference on `Photo` with artificial distribution shifts generated by SGC as the GNN generator.

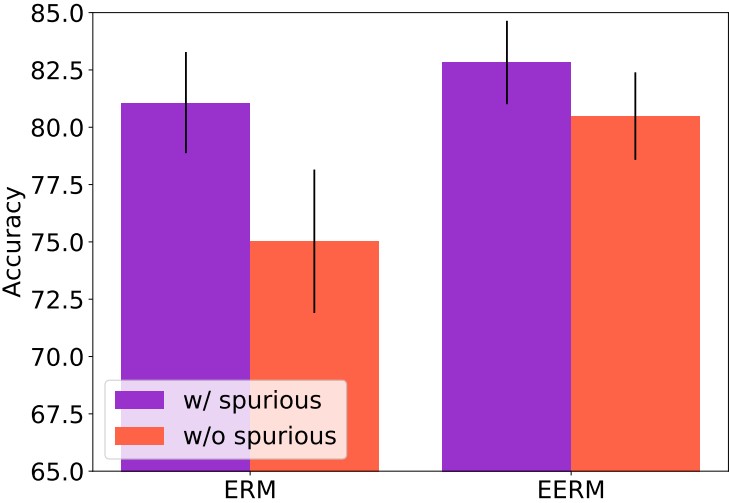

Figure 17: Comparison of training accuracy using all the features v.s. removing the spurious features for inference on `Photo` with artificial distribution shifts generated by GAT as the GNN generator.

