# OpenReview forum: "Handling Distribution Shifts on Graphs: An Invariance Perspective"
_ICLR.cc/2022/Conference — ICLR 2022 Poster_

### Official Review · Reviewer_DyjE · 2021-10-30

**Correctness:** 3
**Technical Novelty And Significance:** 2
**Empirical Novelty And Significance:** 2
**Recommendation:** 5
**Confidence:** 4

**Main Review:**

Strength:
- Provided a new approach to retrieve invariance information among different environments.
- Presented the theoretical foundations and the proposed approach separately in a clear and persuading manner.

Weakness:
- The contribution of adversarial attack or resilient GNN beyond prior work is not fully discussed. Simply list a few at the bottom.
- The OOD problem has been discussed in graph neural network adversarial robustness literature and causal inference literature. Although this paper did a fairly good job in tackling this problem from a different perspective, the novelty is not as great as claimed.
- The experiment only considered one baseline which is ERM. Other competing methods such as structural risk minimization didn't get enough attention. The experimental design looks good otherwise.
- The function in equation 1 doesn't look right. Should be [l(f(x), y)|e].
- Fig 1(b) lack detailed explanation.
- Baselines do not include


- TDGIA: Effective Injection Attacks on Graph Neural Networks. Zou Xu, Zheng Qinkai, Dong Yuxiao, Guan Xinyu, Kharlamov Evgeny, Lu Jialiang, Tang Jie. KDD 2021.
- Adversarial Attacks on Graph Neural Networks via Node Injections: A Hierarchical Reinforcement Learning Approach. Sun Yiwei, Wang Suhang, Tang Xianfeng, Hsieh Tsung-Yu, Honavar Vasant. WWW 2020.
- All You Need Is Low (Rank) Defending Against Adversarial Attacks on Graphs. Entezari Negin, Al-Sayouri Saba A, Darvishzadeh Amirali, Papalexakis Evangelos E. WSDM 2020.
- Gnnguard: Defending graph neural networks against adversarial attacks. Zhang Xiang, Zitnik Marinka. arXiv preprint arXiv:2006.08149 2020.
- Graph Random Neural Networks for Semi-Supervised Learning on Graphs. Feng Wenzheng, Zhang Jie, Dong Yuxiao, Han Yu, Luan Huanbo, Xu Qian, Yang Qiang, Kharlamov Evgeny, Tang Jie. Advances in Neural Information Processing Systems 2020.


**Summary Of The Paper:**

- This paper tackles the out-of-distribution problem for node-level prediction on graphs from the invariance perspective. It presents a novel approach in which the whole graph is divided into n ego-graphs where n is the number of nodes. All the ego-graphs can be treated as a set of IID. Based on this division, the model can be trained to defend the adversarial attack from multiple environments.
- This work extends the discussion of the OOD problem to node-level tasks on graphs. A new learning approach is proposed and theoretically proven to be correct. In empirical experiments on multiple datasets, the approach (EERM) outperforms its counterpart baseline ERM.


**Summary Of The Review:**

I recommend a reject. This paper developed a new approach to tackle a problem that has already been discussed. Although the presentation is well organized and easy to read, the limited novelty and the lack of excitement bring me to this recommendation.

---

> ### Author Response · Authors · 2021-11-13
> **Response to Reviewer DyjE**
>
> Thank you for the time and review. We believe that many of the reviewer's comments/criticisms are predicated on the assumption that existing work on adversarial robustness and our OOD contributions are somehow overlapping.  However, there are actually critical differences such that we believe the novelty of our submission remains intact.  Below, we provide a detailed illustration for our problem setting (with a clear comparison to adversarial robustness) and our technical contributions, which we hope will be useful for evaluation purposes.  We have also included new experiments comparing against adversarial robustness methods applied to the OOD setting.  These make our paper stronger and highlight the superiority of our apprach.  This was a good suggestion.
>
> **Q1**: This paper developed a new approach to tackle a problem that has already been discussed.
>
> **R1**: Actually the problem we address is in fact largely a new one (not something already well-studed in the literature). We focus on OOD generalization for node-level prediction on graphs. As we mentioned in the paper, we are the first to formally present the definition for the problem and provide a systematic framework with theoretical grounded learning approach based on invariant risk minimization.
>
> - ***Novelty*** As far as we know, OOD generalization for node-level problem on graphs remains under-explored (see the comments from Reviewer SVEc) and its formulation is not clearly established in the literature due to the unique challenges (interconnection of nodes and structural information). Also, utilizing invariance principle for handling OOD problem on graphs also remains a new research direction.
>
> - ***Significance and Impact*** Achieving OOD robustness is a classic problem but has nonetheless received much attention recently due to its practical significance. However, handling OOD generalization is challenging and even impossible without any structural assumptions [1,2]. Recent works on invariant risk minimization [3,4,5,6] open a new direction for this problem by resorting to invariant predictive distributions. While a surge of follow-up work has made solid progress in general IID settings, the exploration of invariant learning on graph-structured data has remained under-explored. Our work presents a systematic discussion on applying invariance principles for node-level prediction on graphs, including a new perspective for the data-generating assumption, a new invariant learning method, accompanying theory and newly introduced evaluation protocols. We believe these contributions could impact different communities.
>
> **Q2**: The contribution of adversarial attack or resilient GNNs is not fully discussed.
>
> **R2**: Our problem setting and method are quite distinct from adversarial attacks. Please see the answer to Q3 below for detailed comparison.
>
> **Q3**: The OOD problem has been discussed in GNN adversarial robustness literature and causal inference literature.
>
> **R3**: We need to point out that OOD generalization and adversarial attack/robustness are two different research problems (e.g., see [7,8] for comparison). We compare them in detail.
>
> - ***Comparison with GNN adversarial robustness*** On the problem setting side, adversarial attack methods focus on improving the robustness to noised, contaminated or attacking inputs (e.g. via removing/adding edges or modifying node features) [9,10], while OOD generalization aims to improve the generalization ability of models to out-of-distribution data, i.e., robustness to distribution shifts [11,12]. The essential difference is that for adversarial attack, the contaminated inputs are artificial data rather than real data, while for OOD generalization, the testing OOD data should lie in the manifold generated from a real data distribution [7]. On the methodology side, adversarial attack approaches often focus on designing various data augmentations via introducing random or learnable noise into the training data, while OOD generalization often relies on some structural assumption for data generation (e.g., resort to the invariance in data). More specifically, the key difference of our method is that our objective Eqn. 5 adversarially trains multiple generators to maximize the variance of risks, while most of the methods on adversarial robustness attempt to train an attacker to maximize the loss. Even so, as further support for these claims, we run new experiments using two recent works [9,10] on GNN adversarial robustness as the baselines: Topo Defense and GNNGuard. The results on Twitch dataset with GCNII as the backbone are shown below.
> | Dataset    | ES |  FR | PTBR | RU | TW |
> |------------|-------|-------|-------|-------|-------|
> | Topo Defense  | 63.89 ± 0.91  | 60.89 ± 0.34 | 62.09 ± 0.52 | 55.08 ± 0.28 | 58.21 ± 0.32 |
> | GNNGuard | 63.62 ± 0.89 | 60.14 ± 0.61 | 61.67 ± 0.41 | 54.37 ± 0.56 | 57.78 ± 0.18 |
> | EERM   | 65.54 ± 0.59 | 63.82 ± 0.40 | 63.94 ± 0.48 | 56.21 ± 0.17 | 58.75 ± 0.31 |

---

> > ### Author Response · Authors · 2021-11-13
> > **Response to Reviewer DyjE (Cont.)**
> >
> > - ***Links with causal inference*** Causal inference approaches represent a valid solution for OOD problems. Indeed, the invariant risk minimization or invariant learning can be seen as causal inference method (e.g., see [1,2,3,5] for detailed discussions). Yet, OOD problem for graph-structured data still remains under-explored, especially for node-level prediction on graphs. Therefore, our claims in the original paper are correct.
> >
> > **Q4**: The experiment only considered one baseline ERM.
> >
> > **R4**: We agree that it is always better to consider more comparative methods in experiments. In the answer of Q3, we supplement two works on GNN adversarial attacks as baselines. Also, please see the answer of Q2 for Reviewer SVEc where we add another two baselines (DropEdge and Rex) for comparison. As for the structural risk minimization suggested by the reviewer, can the reviewer describe more precisely which particular method we should consider?  If so, we are happy to include this baseline.
> >
> > **Q5**: The Eqn. 1 does not look right.
> >
> > **R5**: Thanks for pointing out the typo. It should be $l(f(x), y)$.
> >
> > **Q6**: Fig. 1(b) lacks detailed explaination
> >
> > **R6**: This subfigure is an illustration for Assumption 1 which is an extension of the basic invariance assumption [1,2,3] and accommodates the structural information of graph data. As illustrated in the caption of Fig 2 and the paragraph after Assumption 1 (in our paper), we assume that the neighbored nodes in each layer contribute to a portion of invariant predictive information (extracted by $h^*_l$) for the label of centered node $y_v$. The invariant information among neighborhood can be extracted via recursive aggregation along the BFS tree rooted at node $v$, similar to the WL test graph.
> >
> > **Q7**: Baselines do not include
> >
> > **R7**: See answers to Q3 and Q4.
> >
> > **References**
> >
> > [1] Peter Bühlmann. Invariance, causality and robustness. arXiv preprint arXiv:1812.08233, 2018.
> >
> > [2] Mateo Rojas-Carulla, Bernhard Schölkopf, Richard E. Turner, and Jonas Peters. Invariant models for causal transfer learning. Journal of Machine Learning Research, 19:36:1–36:34, 2018.
> >
> > [3] Martín Arjovsky, Léon Bottou, Ishaan Gulrajani, and David Lopez-Paz. Invariant risk minimization. CoRR, abs/1907.02893, 2019.
> >
> > [4] Masanori Koyama and Shoichiro Yamaguchi. When is invariance useful in an out-of-distribution generalization problem ? CoRR, abs/2008.01883, 2021.
> >
> > [5] David Krueger, Ethan Caballero, Jörn-Henrik Jacobsen, Amy Zhang, Jonathan Binas, Rémi Le Priol, and Aaron C. Courville. Out-of-distribution generalization via risk extrapolation (rex). In International Conference on Machine Learning (ICML), 2021.
> >
> > [6] Elliot Creager, Jörn-Henrik Jacobsen, and Richard S. Zemel. Environment inference for invariant learning. In International Conference on Machine Learning (ICML), pp. 2189–2200, 2021.
> >
> > [7] David Stutz, Matthias Hein, Bernt Schiele. Disentangling Adversarial Robustness and Generalization, CVPR 2019.
> >
> > [8] Kimin Lee, Kibok Lee, Honglak Lee, Jinwoo Shin. A Simple Unified Framework for Detecting Out-of-Distribution Samples and Adversarial Attacks, NeurIPS 2018.
> >
> > [9] Xiang Zhang, Marinka Zitnik, GNNGUARD: Defending Graph Neural Networks against Adversarial Attacks, NeurIPS 2020.
> >
> > [10] Kaidi Xu , Hongge Chen, Sijia Liu, Pin-Yu Chen, Tsui-Wei Weng, Mingyi Hong and Xue Lin, Topology Attack and Defense for Graph Neural Networks: An Optimization Perspective, IJCAI 2019.
> >
> > [11] Pang Wei Koh et al. Wilds: A Benchmark of in-the-Wild Distribution Shifts. ICML 2021.
> >
> > [12] Nanyang Ye, Kaican Li, Lanqing Hong, Haoyue Bai, Yiting Chen, Fengwei Zhou, Zhenguo Li. OoD-Bench: Benchmarking and Understanding Out-of-Distribution Generalization Datasets and Algorithms

---

> > > ### Comment · Reviewer_DyjE · 2021-11-29
> > > **after rebuttal**
> > >
> > > I'm not convinced that adversarial attack is entirely distinct from OOD; they may differ yet overlap.
> > >
> > > Good work on adding more baselines and correcting the typo.
> > >
> > > I raise my score to 5 accordingly.

---

> > > > ### Author Response · Authors · 2021-11-30
> > > > **Further response by authors**
> > > >
> > > > Thank you for the time and nice feedback. We agree that OoD generalization and adversarial robustness have some overlaps in terms of formulation. Especially, the latter can be seen as OoD data from a pertubation set.
> > > >
> > > > Yet, apart from the difference in problem-solving (as we discussed in previous response), their difference in evaluation also matters. For adversarial robustness, the evaluation is based on attacking data added with noise or modification to the original inputs. For OoD problem or distribution shift [1], the evaluation focuses on performance on testing data from unseen domains. More specifically, in graph learning tasks, adversarial robustness considers testing data on pertubated graphs (adding/deleting edges or nodes or modifying the features), while OoD generalization deals with graphs from new real environments (e.g., dataset with different structural properties [2,3], new feature subgroups [4] or future observation).
> > > >
> > > > Furthermore, we understand that the reviewer's main concern is that our contribution on problem side might be weakened given the overlaps with adversarial robustness. Yet, our formulation for OoD node-level problem on graphs can essentially be seen as a generalized version for robust learning (the environment can be specified as arbitrary concept, e.g. the pertubation set). Also, our technical contributions (method, theory and evaluation) based on our problem formulation are orthogonal to existing works on robust learning over graphs. We believe our work explores a new aspect for node-level distribution shift problem and could inspire new future research in this direction.
> > > >
> > > >
> > > > [1] Pang Wei Koh et al. Wilds: A Benchmark of in-the-Wild Distribution Shifts. ICML 2021.
> > > >
> > > > [2] Keyulu Xu et al. HOW NEURAL NETWORKS EXTRAPOLATE- FROM FEEDFORWARD TO GRAPH NEURAL NETWORKS. In ICLR 2021.
> > > >
> > > > [3] Beatrice Bevilacqua et al. Size-Invariant Graph Representations for Graph Classiﬁcation Extrapolations. In ICML 2021.
> > > >
> > > > [4] Ma, Jiaqi et al. Subgroup Generalization and Fairness of Graph Neural Networks. NeurIPS 2021.

---

### Official Review · Reviewer_SVEc · 2021-11-02

**Correctness:** 4
**Technical Novelty And Significance:** 3
**Empirical Novelty And Significance:** 3
**Recommendation:** 6
**Confidence:** 2

**Main Review:**

< Strength >
1. OOD generalization for graph neural networks is important, but it is a relatively unexplored research area.
This paper will be attractive for many graph-related machine learning researchers.

2. OOD for graph dataset are not trivial because of their own unique characteristic of the graph-structured dataset (for example, there are only one graph), but the proposed approach, "Explore-to-Extrapolate Risk Minimization",  that adopts K context generators solve the problems in a smart way.

3. The proposed method is validated empirically and theoretically.

< Weakness and Questions >
1. There are some related graph OOD works [1,2]. Additional discussion will be helpful for the audience. Especially, this paper seems similar to [1], and a detailed discussion will be needed.

2. The baseline methods for experimental results are not enough. For example, Table 3 reports the quantitative results for the OGB-Arxiv dataset, but the baseline is only ERM.

3. In [3], they provide the benchmark settings and some results for OGBN-Arxiv. The quantitative results for normal OGBN-Arxiv experimental settings will be interesting.

[1] Bevilacqua, Beatrice, et al. "On Single-environment Extrapolations in Graph Classification and Regression Tasks." (2020).

[2] Bayer, Jens, David Münch, and Michael Arens. "Image-based OoD-Detector Principles on Graph-based Input Data in Human Action Recognition." arXiv preprint arXiv:2003.01719 (2020).

[3] Hu, Weihua, et al. "Open graph benchmark: Datasets for machine learning on graphs." arXiv preprint arXiv:2005.00687 (2020).

**Summary Of The Paper:**

The importance of out-of-domain (OOD) generalization has emerged, and there is much research for out-of-domain generalization [1,2,3,4].
However, the related works on the graph-structured datasets are not explored well.
OOD generalization of the graph is not trivial because the graph has the interconnection among nodes and the existence of structural information.
To solve the problems, this paper proposes a new method, multiple contexts explore that are adversarially trained to maximize the variance of risk.
The proposed model is validated on many diverse datasets, Cora, Amazon-Photo, Twitch-explicit, and so on.

[1] Arjovsky, Martin, et al. "Invariant risk minimization." arXiv preprint arXiv:1907.02893 (2019).

[2] Sagawa, Shiori, et al. "Distributionally robust neural networks for group shifts: On the importance of regularization for worst-case generalization." arXiv preprint arXiv:1911.08731 (2019).

[3] Krueger, David, et al. "Out-of-distribution generalization via risk extrapolation (rex)." International Conference on Machine Learning. PMLR, 2021.

[4] Creager, Elliot, Jörn-Henrik Jacobsen, and Richard Zemel. "Environment inference for invariant learning." International Conference on Machine Learning. PMLR, 2021.

**Summary Of The Review:**

Please see the weakness and questions section.

---

> ### Author Response · Authors · 2021-11-13
> **Response to Reviewer SVEc**
>
> Thank you for the nice comments and valuable suggestions. We are encouraged that you appreciated our technical contributions including the problem significance, novelty, soundness and solid experiments. In the response below, we provide answers to your questions in order to address the concerns and increase your confidence.
>
> **Q1**: Related works
>
> **R1**: Thanks for sharing these two related works. [1] considers invariant learning for extrapolation from a single environment over graph data. Yet it focuses on graph-level prediction problems while we focus on node-level prediction. As we mentioned in Sec 6, the key difference is that for graph-level tasks, each instance is a whole graph (which is often small) and a dataset within one environment can be seen as a set of IID pairs $\{(x_i, y_i)\}$ from $P(x, y | e)$, while for node-level tasks (our focus), each instance is a node in one large graph and instances are inter-connected resulting in a non-IID nature. Such a technical challenge hinders the trivial extension of existing well-established OOD approaches. Our work resolves this issue via decomposing an entire graph into ego-graphs in the generation process. Furthermore, [1] also targets extrapolation from a single environment, which is similar to our case, yet we adopt different methods. [1] achieves this goal via bridging extrapolation with a classifc random graph model while we target the extrapolation via a new invariant learning approach that is agnostic to specific GNN models. In fact, we believe these methodological aspects are orthogonal to each other.
>
> Moreover, [2] proposes a OOD detector for graph-based data in vision via developing a new metric learning approach. Differently, we formulate the OOD problem for graph-structured data through the lens of invariant risk minimization and propose a new theoretical grounded learning approach.
>
>
> **Q2**: More baseline methods
>
> **R2**: We agree that it is always better to consider more comparative methods in experiments. Yet, to our knowledge, we are the first to leverage invariant risk minimization for node-level prediction on graphs and existing related works have distinct technical aspects. For works on invariant learning in general cases, they focus on image data for evaluation instead of graphs. Also most of them cannot be adapted to our case since they assume multiple observed environments in the data. For works on OOD generalization over graphs, most of existing studies  focus on graph-level tasks instead of node-level prediction. They are not applicable in our setting since in their cases, each graph within one environment is treated as IID data (See the answer to Q1 for further comparison). As further investigation, we supplement new experiments and add three comparative methods by modifying the original methods to make them applicable in our case. The results of ROC-AUC on Twitch dataset with GCNII as the GNN backbone are shown below.
>
> - ***ReX + Aug***: We use the objective of risk extrapolation [3] (a recently proposed invariant learning approach) and use randomly adding/deleting adds for generating data from multiple environments;
>
> - ***DropEdge [4]***: randomly remove partial edges for training;
>
> - ***Topo Adv***: We use the optimization-based graph attacking method [5] and train attacker to adversarially maximize the mean loss.
>
> | Test Graphs    | ES |  FR | PTBR | RU | TW |
> |------------|-------|-------|-------|-------|-------|
> | ReX + Aug | 64.37 ± 0.87 | 61.17 ± 0.42 | 62.17 ± 0.42 | 55.09 ± 0.37 | 58.63 ± 0.29 |
> | DropEdge | 64.21 ± 0.61 | 61.52 ± 0.53 | 62.51 ± 0.31 | 55.12 ± 0.32 | 58.51 ± 0.25 |
> | Topo Adv  | 63.89 ± 0.91  | 60.89 ± 0.34 | 62.09 ± 0.52 | 55.08 ± 0.28 | 58.21 ± 0.32 |
> | EERM | 65.54 ± 0.59 | 63.82 ± 0.40 | 63.94 ± 0.48 | 56.21 ± 0.17 | 58.75 ± 0.31 |
>
> As we can see, EERM achieves the best ROC-AUC, which demonstrates the superiority of our approach for OOD generalization.
>
>
> **Q3**: The original setting for OGB-Arxiv
>
> **R3**: Thanks for the suggestion. The original setting for OGB-Arxiv assumes a large time interval for training data and small time gaps between training and testing, i.e., the papers published until 2017 for training, on 2018 for validation and 2019-2020 for testing. Such a setting only requires the model to handle nearly in-distribution generalization. To make it suitable for evaluating OOD generalization in our case, we modify the data splits in our experiment by using much smaller time interval for training (before 2011) and different time gaps for testing (2014-2016, 2016-2018, 2018-2020). Such a setting allows us to better investigate the efficacy of methods with respect to distribution shifts. Indeed, in Table 3 we observe that as the time difference goes large, the performance of both ERM and EERM goes worse, with EERM suffering much less.

---

> > ### Author Response · Authors · 2021-11-13
> > **References**
> >
> > [1] Bevilacqua, Beatrice, et al. "On Single-environment Extrapolations in Graph Classification and Regression Tasks." openreview 2020.
> >
> > [2] Bayer, Jens, David Münch, and Michael Arens. "Image-based OoD-Detector Principles on Graph-based Input Data in Human Action Recognition." arXiv preprint arXiv:2003.01719 (2020).
> >
> > [3] David Krueger, Ethan Caballero, Jörn-Henrik Jacobsen, Amy Zhang, Jonathan Binas, Rémi Le Priol, and Aaron C. Courville. Out-of-distribution generalization via risk extrapolation (rex). In International Conference on Machine Learning (ICML), 2021.
> >
> > [4] Rong Yu et al. DropEdge: Towards Deep Graph Convolutional Networks on Node Classification, ICLR 2020.
> >
> > [5] Kaidi Xu , Hongge Chen, Sijia Liu, Pin-Yu Chen, Tsui-Wei Weng, Mingyi Hong and Xue Lin, Topology Attack and Defense for Graph Neural Networks: An Optimization Perspective, IJCAI 2019.

---

> > > ### Comment · Reviewer_SVEc · 2021-11-18
> > > **Thank you for your response.**
> > >
> > > Thank you for your response.
> > > My concerns are somewhat resolved.
> > > However, I still believe that the original OGBN-Arxiv setting experiments will be needed.
> > > The proposed algorithms are for OOD generalization, and I know that the original OGBN-Arxiv settings are not ideal settings for OOD task.
> > > However, proposed algorithms should work well on normal tasks as well as OOD generalization tasks.
> > > The additional experiments will be interesting for many readers.

---

> > > > ### Author Response · Authors · 2021-11-21
> > > > **Further response**
> > > >
> > > > Thank you for the further comment and the nice suggestion. We agree that it is an interesting scientific inquiry to study the model's performance on in-distribution generalization tasks. We experiment on the original split of ogb-arxiv dataset with GraphSAGE as the backbone. The test accuracy of our approach is $71.58 \pm 0.25$ and ERM yields $71.49 \pm 0.27$. Therefore, our approach does not degrade for normal setting. Furthermore, we provide more experimental results on Twitch and Cora datasets with respect to normal setting (see our response to Q7 of Reviewer xk81), which strengthens the observation that our approach empirically reduces to standard methods when there is no distribution shift. We hope these results help to address your concern. If you have any further question, please let us know.

---

### Official Review · Reviewer_xk81 · 2021-11-02

**Correctness:** 4
**Technical Novelty And Significance:** 3
**Empirical Novelty And Significance:** 3
**Recommendation:** 6
**Confidence:** 4

**Main Review:**

The paper is well written, polished, and easy to follow. The problem that is studied is important and relevant for the community. The proposed approach is well motivated. The theory is sound, however, the technical novelty is limited. The experiments are well executed and mostly convincing, however, an ablation/sensitivity study is missing.

Comments:
- The discussion in terms of a BFS tree rooted at node v is a bit convoluted. Note, the very general and large class of recursive message-passing GNNs [1] subsumes this. Since the message-passing framework is well established in the literature rephrasing some of the discussions and the assumptions (e.g. Assumption 1) in this framework/language will significantly improve readability. At the very least the relation to message-passing GNNs should be discussed and a citation is warranted.
- The theory is sound, however, the technical novelty is limited. Once the input graph has been decomposed into a set of ego-graphs the definitions, formulations and theory are all straightforward adaptions from the respective versions for IID data. Nonetheless, there is some value in doing this.
- It is not clear how much of the apparent benefit is due to the new objective (i.e. the addition of the variance term) vs. the (inherent) benefit of data augmentation. An ablation study should be performed where: (i) first the auxiliary context generators are not trained to maximize the variance (e.g. instead delete/add edges randomly with different probabilities), and (ii) the variance term in the Eq. 5 is removed training only on the K augmented graphs (e.g. again using random context generators). In any case, a comparison with existing data augmentation techniques for graphs (e.g. [2, 3]) is prudent. Similarly, different "ground-truth" environments can be used to train the generators (see question below).
- Relatedly, the sensitivity w.r.t. the hyperparameters should be studied, at least for key hyperparameters such as L, K, and \beta.
- The are only few details provided about the graph generators even though they are an important component of the proposed approach. For example, as far as I can tell there are no constraints on the generated graphs, which can lead to highly unrealistic graphs. Is this the case in practice? Moreover, for the generator the parametrized matrix P_k requires O(N^2) memory which severely limits scalability, how is this issue handled, e.g. for OGB-Arxiv? In addition, using the REINFORCE algorithm to optimize the generator might be suboptimal since the gradients can suffer from high variance. How sensitive are the generators to hyperparameters?
- It is not clear which results are statically significant (especially in Table 2 and 3).

Questions:
- What is the accuracy of EERM when there is no distribution shift? For example if we train a GCN with ERM vs. EERM on Cora using random train/val/test split without introducing an artificial distribution shift how big is the gap in accuracy? In other words, is there a price for using EERM? Relatedly, how well does EERM perform on the standard OGB-Arxiv split and how do these results compared to the OGB leaderboard (using the original transductive setting)?
- Why is the performance on GAT significantly lower compared to GCN and SGC on Figure 2c? Is GAT more sensitive to distributions shift? If yes, can you provide some insight why? If not, is it an orthogonal issue such as not properly tuning the hyperparameters?
- When training with multiple graphs is the assumption that we have a single environment? What happens if we treat the nodes in each graph as belonging to a separate "ground-truth" environment? This would remove the need for context generators and can directly show the benefits of using the variance term in the objective. Similarly, on OGB-Arxiv each year (or set of years) can be considered a separate environment, using multiple for training and one or two for testing.

References:
1. Gilmer et al. "Neural message passing from quantum chemistry"
2. Wang et al. "NodeAug: Semi-Supervised Node Classification with Data Augmentation"
3. Rong et al. "DropEdge: Towards Deep Graph Convolutional Networks on Node Classification"


**Summary Of The Paper:**

The paper adapts a recently proposed invariant risk minimization approach for tackling distribution shift on node-level predictions on graphs. The main idea is to decompose the graph into a set of ego-graphs rooted at each node, thus incorporating structural information. Since the learning objective requires data from different environments they authors introduce auxiliary context generators trained to maximize the variance loss.

**Summary Of The Review:**

The paper is well written, polished, and easy to follow. The problem that is studied is important and relevant for the community. The proposed approach is well motivated. The theory is sound, however, the technical novelty is limited. The experiments are well executed and mostly convincing, however, an ablation/sensitivity study is missing.  I am willing to increase my score if the authors address the issues with the experimental evaluation that I raised in the main review.

---

> ### Author Response · Authors · 2021-11-13
> **Response to Reviewer xk81**
>
> Thank you for the time, thorough comments and nice suggestions. We are pleased that you acknowledged the significance of our focused problem, theoretical soundness and polished presentation. In the following response, we answer your comments/questions point-by-point, clarify our technical contributions and supplement new experiments as suggested to further strengthen our contributions.
>
> **Q1**: The discussions of a BFS tree rooted at v.
>
> **R1**: We agree that our definition in Assumption 1 has a similar form with message-passing GNNs. As we mentioned in the paper, we bridge the recursive computation with WL test [1], which is a classic algorithm and shares the spirit of many off-the-shelf GNNs [2,3]. Yet our Assumption 1 aims to characterize the data-generating distribution while the message-passing rules of GNNs belong to the inductive bias for modeling. Though they share some appearance in the forms, they have distinct targets. Thanks for this suggestion and we will improve the readability in the revision.
>
> **Q2**: The technical novelty for graph decomposition is limited.
>
> **R2**: We agree that the idea of decomposing an entire large graph into pieces of ego-graphs is explored in the literature for e.g., representation or modeling purposes, but its adoption for characterizing a data-generating distirbution is new to our knowledge. Despite its simplicity, such a perspective allows us to modularize the analysis for different centered nodes given the complexity of node interconnections in graph-structured data. This paves the way for problem-solving and analysis on the newly formulated problem setting. On top of this, we propose a theoretically-grounded invariant learning approach, further develop a practical algorithm and prove the validity and generalization capability via analysis (See our general response at the beginning for more discussions). We believe these technical contributions are novel and have potential impact.
>
> **Q3**: Ablation studies.
>
> **R3**: Thanks for the nice suggestions. We conducted new experiments to answer the raised questions. We compare our method EERM with four simplified variants. 1) Random Aug: We use DropEdge [4] (p=0.2) to randomly remove edges and randomly add partial edges.  2) Importance Aug: We use the method [5] to remove/add edges with importance scores. 3) Random Aug w/o Var: We remove the variance term and use the random data augmentation. 4) Importance Aug w/o Var: We remove the variance term and use the importance scores [5] for data augmentation. The results of ROC-AUC on Twitch dataset with GCNII as the backbone are shown below.
>
> | Test Graphs    | ES |  FR | PTBR | RU | TW |
> |------------|-------|-------|-------|-------|-------|
> | Random Aug   | 64.37 ± 0.87 | 61.17 ± 0.42 | 62.17 ± 0.42 | 55.09 ± 0.37 | 58.63 ± 0.29 |
> | Importance Aug | 64.21 ± 0.61 | 61.52 ± 0.53 | 62.51 ± 0.31 | 55.12 ± 0.32 | 58.51 ± 0.25 |
> | Random Aug w/o Var     | 63.52 ± 0.61 | 61.31 ± 0.39 | 61.91 ± 0.24 | 55.25 ± 0.71 |  58.49 ± 0.34  |
> | Importance Aug w/o Var     | 63.63 ± 0.53 | 61.28 ± 0.41 | 62.14 ± 0.29 | 55.42 ± 0.57 |  58.43 ± 0.28  |
> | EERM | 65.54 ± 0.59 | 63.82 ± 0.40 | 63.94 ± 0.48 | 56.21 ± 0.17 | 58.75 ± 0.31 |
>
> The results show that our learnable context generators as well as the variance loss indeed help to enhance OOD generalization performance. For comparison with multi-graph training using different ground-truth environments, please see the answer of Q9.
>
>
> **Q4**: Sensitivity to hyper-parameters.
>
> **R4**: We study the sensitivity of hyper-parameters: number of GNN layers $L$, number of graph generators $K$, weight for variance $\beta$ and sampling size $s$. The results on test graph ES of Twitch dataset with GCNII as the backbone are shown below.
>
> | $L$    | 3 |  7 | 10 | 15 | 20 |
> |------------|-------|-------|-------|-------|-------|
> | ROC-AUC    | 62.92 ± 0.28 | 63.48 ± 0.67 | 63.89 ± 0.46 | 64.07 ± 0.47 | 63.99 ± 0.61 |
> | $K$    | 2 | 3 | 4 | 5 | 6 |
> | ROC-AUC    | 62.91 ± 0.32 | 63.81 ± 0.42  | 64.21 ± 0.32 | 63.78 ± 0.40 | 63.21 ± 0.31  |
> | $\beta$    | 0.1 | 0.5 | 1.0 | 5.0 | 10.0 |
> | ROC-AUC    | 63.106 ± 0.700 | 63.473 ± 0.558 | 63.488 ± 0.671 | 63.530 ± 0.602 | 62.924 ± 0.614 |
> | $s$    | 1 | 3 | 5 | 10 | 20 |
> | ROC-AUC    | 62.81 ± 0.71 | 62.61 ± 0.41 | 62.92 ± 0.45 | 62.76 ± 0.53 | 61.83 ± 0.62 |
>
> Overall, our model is not sensitive to hyper-parameters. The number of GNN layers impacts the model's representation capacity. The number of graph generators would impact how the model explores the environments. Too small $K$ provides limited exploration while too large $K$ may introduce large variance. The $\beta$ controls the balance for optimization on mean and variance losses, so a proper value could bring up optimal performance. Also, when $s$ is small there is no obvious variation in performance and the result becomes worse when it further enlarges (leading to much modification to the graphs).

---

> > ### Author Response · Authors · 2021-11-13
> > **Response to Reviewer xk81 (Cont.)**
> >
> > **Q5**: Graph generators.
> >
> > **R5**: Thanks for raisting these interesting questions. Indeed, for graph generation tasks, adding constraints to the generated graphs is important and brings some difficulty for optimization. In our case, we can properly set the sampling size $s$ to control how much we edit the graph structures in order to accommodate the constraints through learning. Empirically, we set $s$ less than 5 and found it works smoothly. Furthermore, the scalability issue is indeed a limitation for our current method. As far as we can tell, it is indeed a technical challenge for graph structure learning [6,7,8] that considers modifying or generating graph structures for optimal message passing. We can use the anchor-based method [8] or neighbor sampling strategy [9] for reducing the complexity, and we leave the exploration for this point as future research. Moreover, we agree that the REINFORCE algorithm may cause large gradient variance in theory. Yet in our case we do not strictly require the graph generators to be optimal and instead we consider iterative training for the graph generators and GNN models in practice (i.e., Alg. 1). Also the graph generators aim to explore the environments instead of fitting the data. We found using REINFORCE works smoothly in practice. For hyper-parameter sensitivity, please see the answer of Q4.
> >
> > **Q6**: Statistical significance.
> >
> > **R6**:  For Tables 2 and 3, most of the improvements are significant with 95% confidence level under Wilcoxon signed-rank test. In the revised pdf, we mark the results that are not statistically significant (only one case in Table 2 and one case in Table 3).
> >
> > **Q7**: What is the accuracy of EERM when there is no distribution shift?
> >
> > **R7**: This is an interesting scientific inquiry and we added new experiments to study it. For the Twitch dataset, we consider a situation without distribution shift: we train on a random set of nodes in DE and test on the remaining nodes in DE. With GCNII as backbone, EERM yields ROC-AUC $70.92 \pm 0.48$ and ERM yields $70.53 \pm 0.43$. Also, for Cora experiments, we use the standard semi-supervised setting (randomly selected 140/500 nodes for training/validation and the remaining for testing), EERM with GCN as backbone yields test accuracy $81.74 \pm 1.23$ and ERM yields $81.24 \pm 1.41$. We can observe that EERM is no worse than ERM, which implies that our approach is stable when there are no OOD shifts.
> > Such a phenomenon also conforms with OGB-Arxiv experiment in Table 3 where we evaluate the trained model on testing data within different time intervals. When the time gap is small (distribution shift is small), EERM yields similar performance to ERM or slightly better. As we enlarge the time differences, the distribution shifts become more significant (verified by the visualization in Fig. 9), and the performance gain of EERM over ERM becomes larger as well.
> >
> > **Q8**: Why is the performance on GAT significantly lower compared to GCN and SGC in Fig. 2c?
> >
> > **R8**: We think there might be a misunderstanding regarding our experiments in Fig. 2c. Here we consider GCN/SGC/GAT as the GNN generating input data with artificial distribution shifts (see the third paragraph in Sec 5.3 for illustration and Appendix E.1 for details). The GNN backbone for the prediction model is set as GCN in all the cases. Therefore, the performance drop of GAT in Fig. 2c is mainly because the GCN backbone has insufficient capacity for modeling data generated from GAT. We will add more descriptions in the revision for clearer presentation.
> >
> > **Q9**: Training with multiple graphs and each graph as a ground-truth environment.
> >
> > **R9**: This is an insightful question. In fact, in our experiments on FB datasets, we consider using multiple graphs for training and we indeed treat each graph as a unique environment. More specifically, we use three graphs for training and the graph generators in our model will yield 3*K augmented graphs to calculate the mean and variance losses. As further investigation, we remove the graph generators in our model and directly train the model with three training graphs each of which is assigned a unique environment id. When using John Hopkins + Caltech + Amherst as training graphs, we obtain the test accuracies of Penn, Brown, Texas as $50.15 \pm 0.25$, $55.73 \pm 0.42$, $52.31 \pm 1.04$, which are significantly worse than EERM which gives $50.69 \pm 0.30$, $56.59 \pm 0.19$, $54.86 \pm 1.39$. The results suggest that our graph generators indeed help to explore environments and benefit OOD generalization.

---

> > > ### Author Response · Authors · 2021-11-13
> > > **References**
> > >
> > > [1] Boris Weisfeiler and AA Lehman. A reduction of a graph to a canonical form and an algebra arising during this reduction. Nauchno-Technicheskaya Informatsia, 2(9):12–16, 1968.
> > >
> > > [2] Keyulu Xu et al.. How powerful are graph neural networks? ICLR 2020.
> > >
> > > [3] Zhengdao Chen et al. On the equivalence between graph isomorphism testing and function approximation with GNNs. arXiv:1905.12560v1, 2020.
> > >
> > > [4] Yiwei Wang et al. NodeAug: Semi-Supervised Node Classification with Data Augmentation, KDD 2020.
> > >
> > > [5] Rong Yu et al. DropEdge: Towards Deep Graph Convolutional Networks on Node Classification, ICLR 2020.
> > >
> > > [6] Luca Franceschi et al. Learning Discrete Structures for Graph Neural Networks. ICML 2019.
> > >
> > > [7] Wei Jin et al. Graph Structure Learning for Robust Graph Neural Networks. KDD 2020.
> > >
> > > [8] Yu Chen et al. Iterative Deep Graph Learning for Graph Neural Networks: Better and Robust Node Embeddings. NeurIPS 2020.
> > >
> > > [9] William L. Hamilton et al. Inductive Representation Learning on Large Graphs, NeurIPS 2017.

---

> > > > ### Comment · Reviewer_xk81 · 2021-11-29
> > > > **Raising the score**
> > > >
> > > > I thank the authors for the details response, the clarifications, and the additional experiments. After taking their response into account, as well as the rest of the reviews and discussions I am raising my score to 6.

---

### Official Review · Reviewer_Wo9P · 2021-11-03

**Correctness:** 3
**Technical Novelty And Significance:** 2
**Empirical Novelty And Significance:** Not applicable
**Recommendation:** 6
**Confidence:** 3

**Main Review:**

## Strength
1. **Overall, the idea is new to me and experiments are comprehensive.**
2. **The theorem proposed are rigorously proved in the appendix.**

## Weaknesses
1. **Missing some related work.**
Besides OOD on graph and other domain, there are couple of work [1,2] study the generalization from non-I.I.D training data in GNNs. Some discussion on this is necessary.
1. **Technical writing is dense and hard to follow**
In Figure 1, the context generator is actually not generating a new graph, which instead augment the input graph. The terminology used here is confusing.

1. **Assumption 1 seems to be unrealistic.**
In practice, do we really have the invariance property? Any kind of study supports this. I feel this assumption is the foundation of the paper that tries to bridge OOD and invariant risk minimization. For example, after graph editing, it would be hard to obtain invariant property. If we remove several author and add authors from different domain, the paper category will probably change.

[1] Ma, Jiaqi, Junwei Deng, and Qiaozhu Mei. "Subgroup Generalization and Fairness of Graph Neural Networks." NeurIPS 2021.

[2] Zhu, Qi, et al. "Shift-Robust GNNs: Overcoming the Limitations of Localized Graph Training Data." NeurIPS 2021.

**Summary Of The Paper:**

This paper studies the problem of distribution shifts as out-of-distribution generalization. Specifically, it formulates the OOD problem as invariant risk minimization under different environments. The relation between these two has been extensively discussed in the paper. Multiple environment is done by graph editing using policy gradient. Experiments on three different kind of distribution shifts are presented and the effectiveness of EERM is validated.

**Summary Of The Review:**

After reading this paper several times, I start to understand the underlying connection between OOD and invariant property. I think the readability of the paper can be greatly improved by introducing the connection first then talk about their implementation. However, the creation of multiple environment while keeping the invariant property is unclear to me. I might misunderstand this part. Currently, I think it is marginally below the acceptance.

---

> ### Author Response · Authors · 2021-11-13
> **Response to Reviewer Wo9P**
>
> Thank you for the time and thorough reviews. We are glad that you liked our method, theory and experiments. Here are our responses to your questions:
>
> **Q1**: Missing some related works.
>
> **R1**: Thanks for sharing with us these up-to-date works; we were not previously aware of them as they were just accepted by NeurIPS'21, with a notification time that is after the ICLR submission deadline. [1] focuses on subgroup generalization and presents a PAC-Bayesian theoretic framework, while [2] targets distribution shifts between selecting training and testing nodes and proposes new GNN model called Shift-Robust GNNs. While our work is related, the key differences are three-fold. 1) We focus on general OOD generalization problems, where distribution shifts between training and testing data can stem from arbitrary causes, and formulate such a problem in a formal way (Eqn. 2) in the context of node-level prediction on graphs. 2) We investigate the OOD generalization over graphs through the lens of invariant risk minimization and propose a new invariant learning approach with theoretical guarantees. 3) We experiment on three scenarios with distinct distribution shifts and test our method (that is model-agnostic) with various GNN backbones. Nonetheless, we will cite these works and add relevant discussion in the revised version.
>
> **Q2**: Technical writing is dense and hard to follow.
>
> **R2**: We are sorry for causing some unnecessary difficulty for readers, although the page limit for the main text makes it somewhat challenging to include both the necessary technical precision and more intuitive companion explanations. We also understand that the concepts and techniques of invariant learning are new for graph learning, even if the research on invariant learning for general OOD problems (e.g., images) has been extensively studied [3,4,5,6]. Also, thank you for pointing out the confusing part of Fig. 1; and we will change "generating" to "editing" in the revision, good suggestion.
>
> **Q3**: Assumption 1 seems to be unrealistic.
>
> **R3**: We believe that assumption 1 is actually quite realistic per the following line of reasoning.  The connection between invariance properties and OOD problems has been extensively studied within the framework of invariant risk minimization [5] as a new paradigm for handling OOD generalization. The invariance assumption has been introduced by [3,4] and applied by follow-up works [5,6,7,8,9] (e.g. see Sec 2.1 of [6]) as a cornerstone hypothesis for data generation which enables reasonable problem-solving and analysis for OOD problems. In the context of image learning, a common illustrative example is a horse on grassy or sandy filed. The horse object is the invariant part since the relation from horse object to the label is stable across arbitrary environments (where horses stand on various fields). By contrast, the background is the variant feature that contributes to predictions that are sensitive to environment changes. Then the model (e.g., an image classifier trained on dataset with grassy background) could be guided to capture the invariant features through training in order to generalize to new data within new domains (e.g., sandy background).
>
> We also emphasize that the notion of invariance property/assumption for invariant risk minimization or invariant learning [3-9] (i.e., what we focus on) means that the predictor model should be invariant, i.e., the predictive distribution for labels conditioned on a model's representation of inputs should yield equally optimal performance across environments. Such a definition is different from invariant representation learning in multi-view learning [10] or self-supervised learning [11] where the "invariant" means a portion of entries in representation vectors that are shared across the objects from different views (See Sec 2.2 of [7] for more illustration).
>
> Furthermore, we can use an analogy to better understand invariance properties in the context of graph data. For node classification, the distributions of a node's neighbors can be different across domains. In this regard, there could exist a set of neighboring nodes that contribute to causal relations to a target node's label and also another set of neighbored nodes that have non-causal relations. The relation from causal neighbors to labels is stable and invariant across domains while that for non-causal neighbors are variant. An illustrative example comes from citation networks, where the goal is to predict a paper's sub-area and the published venues can be seen as domains or environments. Then there exists some cited papers that have close links to the target paper from technical aspects that contribute to invariant predictive information. But there could also be some cited papers used for informing future directions or other broader context. These citations would vary depending on different published venues. The model is expected to focus on invariant information for prediction.

---

> > ### Author Response · Authors · 2021-11-13
> > **Response to Reviewer Wo9P (Cont.)**
> >
> > Our graph editing can change the distributions for input data, which helps to simulate observed data from multiple environments. Again notice that the invariant property does not require us to obtain inputs with shared features from multiple views (as done in invariant representations), and instead we aim to obtain datasets with different distributions and share an invariant predictive distribution. This requirement is not strict and can be easily achieved though our data augmentation.
> >
> > [1] Ma, Jiaqi, Junwei Deng, and Qiaozhu Mei. "Subgroup Generalization and Fairness of Graph Neural Networks." NeurIPS 2021.
> >
> > [2] Zhu, Qi, et al. "Shift-Robust GNNs: Overcoming the Limitations of Localized Graph Training Data." NeurIPS 2021.
> >
> > [3] Peter Bühlmann. Invariance, causality and robustness. arXiv preprint arXiv:1812.08233, 2018.
> >
> > [4] Mateo Rojas-Carulla, Bernhard Schölkopf, Richard E. Turner, and Jonas Peters. Invariant models for causal transfer learning. Journal of Machine Learning Research, 19:36:1–36:34, 2018.
> >
> > [5] Martín Arjovsky, Léon Bottou, Ishaan Gulrajani, and David Lopez-Paz. Invariant risk minimization. CoRR, abs/1907.02893, 2019.
> >
> > [6] Masanori Koyama and Shoichiro Yamaguchi. When is invariance useful in an out-of-distribution generalization problem ? CoRR, abs/2008.01883, 2021.
> >
> > [7] David Krueger, Ethan Caballero, Jörn-Henrik Jacobsen, Amy Zhang, Jonathan Binas, Rémi Le Priol, and Aaron C. Courville. Out-of-distribution generalization via risk extrapolation (rex). In International Conference on Machine Learning (ICML), 2021.
> >
> > [8] Elliot Creager, Jörn-Henrik Jacobsen, and Richard S. Zemel. Environment inference for invariant learning. In International Conference on Machine Learning (ICML), pp. 2189–2200, 2021.
> >
> > [9] Jiashuo Liu, Zheyuan Hu, Peng Cui, Bo Li, and Zheyan Shen. Heterogeneous risk minimization. In International Conference on Machine Learning (ICML), pp. 6804–6814, 2021.
> >
> > [10] Marco Federici, Anjan Dutta, Patrick Forré, Nate Kushman, and Zeynep Akata. Learning robust representations via multi-view information bottleneck. In ICLR, 2020.
> >
> > [11] Yao-Hung Hubert Tsai, Yue Wu, Ruslan Salakhutdinov, and Louis-Philippe Morency. Self-supervised learning from a multi-view perspective. In ICLR, 2021.

---

> > > ### Comment · Reviewer_Wo9P · 2021-11-29
> > > **Post rebuttal response**
> > >
> > > I thank the reviewer for detailed explanation and I am happy about the added clarifications. I raise my score to 6 accordingly. I agree that invariant risk minimization is relatively to the graph learning community and more introduction on this would improve the paper's readability.

---

### Author Response · Authors · 2021-11-13
**General Response by Authors**

Dear area chair and reviewers,

We appreciate the reviewers' time and valuable comments. Overall, the reviewers thought our work well motivated (xk81, SVEc) and written (xk81, DyjE), and acknowledged our novelty (Wo9P, SVEc), theoretical soundness (Wo9P, xk81, SVEc, DyjE) and comprehensive evaluations (Wo9P, SVEc). The major concerns lie in our invariance assumption (Wo9P), technical novelty (xk81, DyjE) and additional experiments asked by xk81, SVEc, DyjE. However, to clarify upfront some potential misunderstandings that may influence how our work is interpreted, we first restate our contributions and resolve some big picture issues.

- **Problem and Aspect** (See our Section 2) We formally present the definition of out-of-distribution generalization for node-level prediction problem over graphs. As far as we know, despite the fact that the OOD problem has been extensively studied, its formulation for graph data and especially node-level prediction problems still remains unclear. And critically, node-level problems on graphs involve the interconnection among data nodes, which results in the non-IID nature of data within one environment and hinders the trivial extension of existing well-established formulation.

- **Methodology** (See our Section 3) We attempt to solve the problem via invariance principles and propose a new invariant learning approach, explore-to-extrapolate risk minimization, that can achieve extrapolation from a single observed environment (a common case for node-level problems). Such a merit clearly distinguishes our method from existing invariant learning approaches in general settings. Also, to our knowledge, we are the first to leverage invariant learning for node-level prediction on graphs.

- **Theory** (See our Section 4) We provide rigorous theoretical results to back up our proposed method, providing guarantees of a valid OOD solution and revealing its effect on tightening an information-theoretic upper bound of the generalization error.

- **Evaluation** (See our Section 5) We design comprehensive experiments, including three different distribution shifts on real-world datasets, to evaluate our method and show its superiority over empirical risk minimization with various off-the-shelf GNN backbones. Also, our experimental designs could serve to enrich the benchmarking zoo for OOD problems recently coming to the spotlight.

In the following individual response, we provide detailed answers to all the questions/comments point-by-point and also supplement new experimental results to further strengthen our contributions.

---

### Decision · Program_Chairs · 2022-01-20

**Decision:**

Accept (Poster)

**Comment:**

The reviewers have improved their scores after the rebuttal, and I agree that the work has value. It proposes a model-driven data augmentation approach to environment-invariant graph representation. Just like most data augmentation works in graph representation learning, the approach relies on graph proposal generator. The work has an implicit mechanism hidden in the graph generator (the authors' reply to a reviewer that "we target the extrapolation via a new invariant learning approach that is agnostic to specific GNN models" is a misunderstanding of why & how these types of methods work). The submission could significantly improve the introduction by properly describing the work w.r.t. other OOD efforts in graph representation learning. While the tasks of different works may be different (e.g., graph classification vs node classification), it is important to emphasize what each different approach brings to the table (rather than just state that the tasks are different). I hope the authors take this opportunity to improve the introduction. This work is more similar to the former OOD graph representation works than IRM & REX, since (without modeling assumptions) both IRM and REX require the support of the environments in test to be a subset of the ones in training. The present work does not need this support assumption since it uses an implicit mechanism in the graph generator.

The toy example in Section 3.1 is informative, thank you. The theory looks solid and easy to understand. The technical novelty is limited, since once the input graph has been decomposed into a set of ego-graphs the definitions, formulations, and theory are all straightforward adaptations from the respective versions for IID data.

Minor:
- In Assumption 2, m() should be defined inside the assumption.